# Projection-Enhanced Contrastive Learning and Linear Calibration for Exemplar-Free Class-Incremental Learning

## Abstract

Exemplar-Free Class-Incremental Learning (EFCIL) tackles the challenge of learning to discriminate between new and old classes without retaining any past exemplars. Contrastive learning offers a promising direction to mitigate feature-space congestion in EFCIL, yet applying it directly in the classification feature space interferes with the objective of learning class-discriminative features, harming performance. We thus propose a novel EFCIL framework decoupling the contrastive learning and classification via a projection head in order to take advantage of the contrastive learning and preserve rich class-discriminative features in the pre-projection space. To further reduce congestion between old and new classes, we propose an old-class repulsion strategy directly on the pre-projection space. Additionally, we propose to eliminate the computation overhead incurred by current prototype calibration methods through a closed-form similarity-weighted linear regression update, enabling efficient yet effective adaptation of full prototype distributions. By integrating these three strategies, our proposed method outperforms existing state-of-the-art methods across several benchmarks. Code available at https://anonymous.4open.science/r/iclr2026-D134.

## 1 Introduction

Deep neural networks have achieved remarkable success across diverse domains, but their results typically rely on stationary data distributions and access to all training data at once. In real-world scenarios, data often arrive incrementally over time, and simply fine-tuning on new data leads to catastrophic forgetting (McCloskey & Cohen, 1989; French, 1999). To address this issue, continual learning methods have emerged (Parisi et al., 2019; De Lange et al., 2022; Zhou et al., 2024), among which exemplar replay, retaining a small buffer of past data, remains dominant, especially in class-incremental learning (CIL) (Rebuffi et al., 2017; Chaudhry et al., 2019; Zhou et al., 2024). Despite its effectiveness, in privacy-sensitive domains such as healthcare, storing data of past tasks is prohibited. This restriction has motivated EFCIL methods (Shi & Ye, 2024; Rypeść et al., 2024b).

By performing classification using Mahalanobis distance, recent methods (Goswami et al., 2023; Rypeść et al., 2024b) achieve superior EFCIL performances. However, a critical challenge hindering their capability is feature space congestion. Namely, when new tasks are introduced, the features of new classes can overlap with one another as well as with old class features, degrading classification performance for both new and old tasks. Prototype augmentation approaches (Zhu et al., 2021b;a; 2022; Shi & Ye, 2023; Malepathirana et al., 2023) show promise to mitigate the issue, among which FCS takes a further step by proposing a prototype-guided contrastive loss (Li et al., 2024). Yet, due to the augmentation-invariant constraint of the contrastive loss, applying it in the same space as the classification loss tends to overly squeeze the feature space and lead to the loss of informative features for classification. This is particularly harmful for cold-start EFCIL tasks.

Another challenge is to prevent stored old-task prototypes from being outdated as the feature extractor evolves to adapt to new tasks. Prototype calibration methods (Gomez-Villa et al., 2024; Li et al., 2024; Magistri et al., 2024; Rypeść et al., 2024b) use new-task samples as a proxy for old classes and estimate prototype drifts by measuring the difference between the features of the new-task samples extracted by the old and the evolving feature extractors. One line of prototype calibration methods

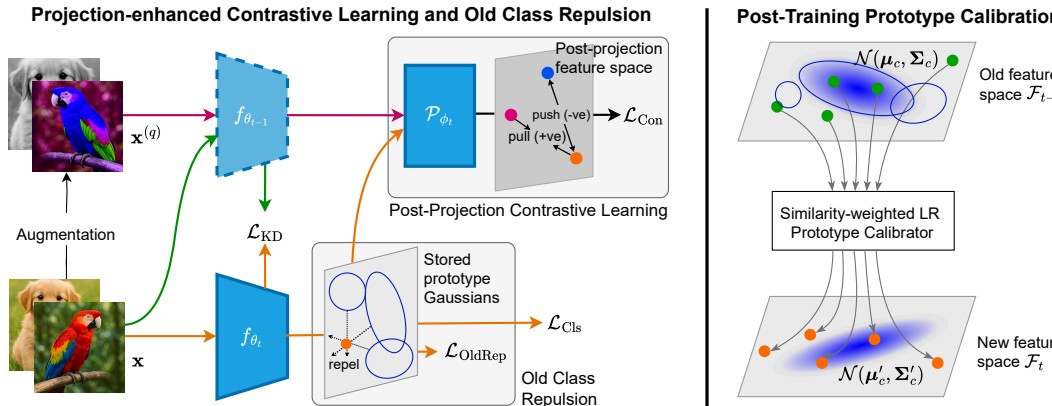

Figure 1: Overview of our EFCIL method. (*left*) Projection head $\mathcal{P}_{\phi_t}$ decouples the contrastive loss $\mathcal{L}_{\text{Con}}$ and the classification loss $\mathcal{L}_{\text{Cls}}$. The projection-enhanced contrastive learning yields richer features by pulling features from the same class features closer and pushing away new class features from different new and old classes in the post-projection space. The old-class repulsion loss $\mathcal{L}_{\text{OldRep}}$ further pushes the new-class samples away from the stored prototype distributions in the pre-projection space without compressing the class features. (*right*) For every old class $c$, we compute the proximity-based similarities between the stored prototype and new-task samples in the old feature space. Then, using these similarities, we formulate a closed-form linear regression solution to map the old features to the new, allowing for efficient Gaussian calibration.

(Yu et al., 2020; Wang et al., 2023; Magistri et al., 2024) estimates the prototype drift by averaging the drift of new task samples, weighted by the proximity of the prototypes to the new task samples in the old feature space. Although they perform well when only the class means are calibrated, updating full prototype distributions becomes time-consuming, as it requires calibrating a large number of samples from the high-dimensional prototype distributions (Rypeść et al., 2024b). Alternatively, (Gomez-Villa et al., 2024; Rypeść et al., 2024b) propose to learn a direct mapping from old to new task feature spaces via a multi-layer perceptron (MLP). Although these approaches speed up the distribution-level updates, they incur significant computational overhead due to the MLP training.

To jointly tackle both feature-space congestion and prototype calibration efficiency, we propose a unified framework. First, to prevent contrastive loss from suppressing classification features as a result of augmentation invariance, we introduce a nonlinear projection head (Xue et al., 2024) that decouples contrastive learning from the classification objective. Specifically, it allows us to apply the classification loss to the pre-projection feature space while performing contrastive learning in the post-projection space simultaneously, as illustrated in Figure 1. By leveraging the projection head filtering out the information irrelevant to the contrastive objective (Ouyang et al., 2025), our proposed method learns richer class-discriminative features in the pre-projection space. Second, we introduce an old-class repulsion loss that repels new-task features away from old-class prototypes in the pre-projection space, thus further reducing congestion between old and new task features to improve classification results. Third, we propose replacing the MLP-based calibration with a closed-form linear-regression-based prototype update without any iterative gradient updates. Furthermore, integrating the similarity between old prototypes and new task features in the least-squares solution allows us to efficiently calibrate each old class prototype distribution.

In summary, our contributions include:

- A projection-enhanced contrastive learning approach along with old-class repulsion strategy to reduce feature space congestion while learning rich class-discriminative features.

- A closed-form similarity-weighted linear-regression-based prototype calibration that allows swift and effective full Gaussian calibration.

- A unified framework that combines these components to achieve superior performance over state-of-the-art (SOTA) methods across multiple EFCIL benchmarks.

- An extensive ablation study validating the benefits of projection-enhanced contrastive learning, old-class repulsion, and our calibration method.

## 2 RELATED WORKS

### 2.1 CLASS-INCREMENTAL LEARNING

CIL does not allow task identity during prediction, and exemplar-replay is the most common approach in CIL, which stores part of past task data to be replayed later (Rebuffi et al., 2017; Chaudhry et al., 2019; Hou et al., 2019; Douillard et al., 2020; Ahn et al., 2021; Caccia et al., 2021; Kang et al., 2022). Other strategies include dynamically expanding the network across tasks (Yan et al., 2021; Zhou et al., 2023), using replay samples to rebalance batch-normalization statistics (Cha et al., 2023; Lyu et al., 2023), and optimizing exemplar selection strategy (Zhou et al., 2022; Hao et al., 2023). Regularization-based methods—originally designed for task-incremental learning, like (Zenke et al., 2017; Kirkpatrick et al., 2016; Nguyen et al., 2018)—can be adapted to CIL by adding task identifiers (Kim et al., 2022). Despite their success, replay-based methods are unsuitable for privacy-sensitive domains that forbid storing past data, motivating exemplar-free alternatives.

### 2.2 EXEMPLAR-FREE CLASS INCREMENTAL LEARNING

There are two broad EFCIL families: *frozen-feature-extractor* (classifier-incremental learning) methods and *evolving-feature-extractor* methods. Frozen-feature-extractor methods (Goswami et al., 2023; Petit et al., 2023; Ma et al., 2023; Zhuang et al., 2024) train the feature extractor on the first task and keep it frozen during the subsequent tasks. Despite their limited plasticity, they perform well in *warm-start* scenarios, where the first task includes nearly 50% of the classes and later tasks are relatively small. However, they underperform in challenging *cold-start* scenarios due to the limited feature representation learned from a small first task. To address this limitation, several evolving-feature-extractor methods have been proposed (Rypeść et al., 2024b; Magistri et al., 2024; Gomez-Villa et al., 2024). Our work follows this second, evolving extractor line of research.

**Evolving feature extractor and prototype calibration** LwF-MC, introduced in (Rebuffi et al., 2017) as a variant of (Li & Hoiem, 2017), is an exemplar-free method that evolves the feature extractor using logit distillation. Subsequent works (Zhu et al., 2021a;b; 2022; Shi & Ye, 2023; Malepathirana et al., 2023) improve EFCIL performance by combining feature distillation with prototypical network and different *prototype augmentation* strategies for robust representation of past-task features. However, these methods face a common limitation: the stored prototypes become outdated as the feature extractor adapts to new tasks, degrading accuracy. To address this limitation, *prototype calibration* methods map the saved old class prototypes to the updated feature space. Some estimate prototype drift with a weighted average of new task samples' drift, weighted by the proximity of the prototypes to the new-task samples in the old feature space (Yu et al., 2020; Wang et al., 2023; Magistri et al., 2024). Others learn a direct mapping from the old to the new feature space using an MLP via stochastic gradient updates (Gomez-Villa et al., 2024; Li et al., 2024; Rypeść et al., 2024b). Instead, we propose a similarity-weighted least-squares solution as a simpler and more efficient alternative to these stochastic gradient updates.

**Reducing feature congestion between old and new classes** PRL (Shi & Ye, 2024) encourages new class features to be orthogonal to the old prototypes. FCS (Li et al., 2024) then applies prototype-guided contrastive learning to further separate old and new class features. However, the augmentation-invariant constraint inherent to contrastive learning can lead to the loss of features important for classification, resulting in sub-optimal performance. We, thus, propose to decouple the classification objective from contrastive learning through a projection head and also encourage knowledge distillation by using the old feature extractor as the encoder model for the augmented input views. Likewise, AdaGauss (Rypeść et al., 2024b) and related methods (Goswami et al., 2023; Rypeść et al., 2024a) store the Gaussian distribution for each class as prototypes and use Mahalanobis distance, achieving SOTA results. Although AdaGauss evolves these saved prototypes, it does not explicitly mitigate the interference between old and new classes. In contrast, our projection-head-aided contrastive learning, combined with old class repulsion loss, alleviates this congestion and yields superior performance.

## 2.3 SELF-SUPERVISED LEARNING

Several self-supervised learning (SSL) methods (Chen et al., 2020; Zbontar et al., 2021) attach a projection head during representation learning to improve downstream-task performance; the head is then discarded during finetuning. Recent theoretical analyses (Xue et al., 2024; Ma et al., 2024; Ouyang et al., 2025) suggest that enforcing invariance to data transformations via contrastive learning can discard critical features for classification when the contrastive learning is applied on the same feature space as the classification loss. In contrast, with the projection head, the pre-projection space retains richer information. Unlike the standard two-phase "pretrain-then-classify" pipeline, our approach integrates projection-head-aided contrastive learning and the classification objective concurrently in the EFCIL setting.

## 3 METHODOLOGY

### 3.1 PROBLEM STATEMENT

CIL addresses the learning on incremental sequence of $T$ tasks with datasets $\mathcal{D} = \{\mathcal{D}_1, \ldots, \mathcal{D}_T\}$. Each dataset $\mathcal{D}_t$ consists of data points $\{(\mathbf{x}_i, y_i) | y_i \in \mathcal{C}_t\}$, where $\mathcal{C}_t$ represents the label set for task $t$. In CIL, there is no overlap in labels between the tasks, i.e., $\mathcal{C}_i \cap \mathcal{C}_j = \varnothing, \forall i \neq j$. During prediction, the task identity is unknown, requiring prediction over all seen classes $\mathcal{C}_{1:t} = \bigcup_{k=1}^{t} \mathcal{C}_k$. The EFCIL variant prohibits storing past exemplars, making the problem more challenging as the past data cannot be revisited. We consider the offline setting - allowing multiple iterative updates over $\mathcal{D}_t$ and we initialize the feature extractor for the first task from scratch without relying on a pre-trained model.

### 3.2 BASELINE

We adopt AdaGauss (Rypeść et al., 2024b) as our baseline. For each task $t$, the model consists of feature extractor $f(\cdot; \theta_t) \colon \mathbf{x} \mapsto \mathbb{R}^M$, a softmax classifier $g_{\varphi_t}$ and a knowledge distillation projector $\mathcal{T}_{\psi_t}$. The classifier predicts only over classes $\mathcal{C}_t$, and the training minimizes the loss

$$\mathcal{L}_t = \mathcal{L}_{\text{CE}}(\theta_t, \varphi_t; \mathcal{D}_t) + \mathcal{L}_{\text{AC}}(\theta_t; \mathcal{D}_t) + \mathbb{1}_{t>1}\gamma\, \mathcal{L}_{\text{KD}}(\theta_t, \psi_t; \mathcal{D}_t, \theta_{t-1}) \tag{1}$$

where,

$$\mathcal{L}_{\text{CE}} = -\mathop{\mathbb{E}}_{(\mathbf{x}_i, y_i) \sim D_t} [y_i \log (g_{\varphi_t} \circ f_{\theta_t}(\mathbf{x}_i))]$$

is the standard cross-entropy loss and

$$\mathcal{L}_{\text{KD}} = \mathop{\mathbb{E}}_{(\mathbf{x}_i, y_i) \sim D_t} \|\mathcal{T}_{\psi_t} \circ f_{\theta_t}(\mathbf{x}) - f_{\theta_{t-1}}(\mathbf{x})\|^2 \tag{2}$$

denotes the feature-distillation loss, which is applied on tasks $t > 1$ to maintain stability. For methods such as PASS (Zhu et al., 2021b) that omit the KD projector, $\mathcal{T}_{\psi_t}$ can be defined as an identity mapping function. $\mathcal{L}_{\text{AC}}$ represents the anti-collapse loss in AdaGauss, which prevents each class's feature distribution from collapsing, leading to improved classification performance. For convenience, we jointly represent the classification loss as $\mathcal{L}_{\text{Cls}} = \mathcal{L}_{\text{CE}} + \mathcal{L}_{\text{AC}}$.

After training on task $t$, new class prototype mean $\boldsymbol{\mu}_c$ and covariance $\boldsymbol{\Sigma}_c$ are estimated using the trained feature extractor $f_{\theta_t}$. To calibrate the old prototypes $\{(\boldsymbol{\mu}_k, \boldsymbol{\Sigma}_k) | k \in \mathcal{C}_{1:t-1}\}$ with the new feature space, a learnable calibrator $\mathcal{T}_\zeta$ is trained as:

$$\zeta_t = \arg\min_{\mathcal{T}} \mathop{\mathbb{E}}_{(\mathbf{x}_i, y_i) \sim D_t} \|f_{\theta_t}(\mathbf{x}) - \mathcal{T}_\zeta \circ f_{\theta_{t-1}}(\mathbf{x})\|^2 + \mathcal{L}_{\text{AC}}(\zeta; \mathcal{D}_t, \theta_t) \tag{3}$$

Then, for each stored class $c$, multiple samples are drawn from the Gaussian distributions and passed through $\mathcal{T}_{\zeta_t}$, and $(\boldsymbol{\mu}_c, \boldsymbol{\Sigma}_c)$ are re-computed using the transformed samples. The original multi-layer calibrator $\mathcal{T}_{\zeta_t}$, and incorporation of $\mathcal{L}_{\text{AC}}$ requires iterative stochastic-gradient updates to learn the calibrator, which incurs significant overhead. To reduce complexity, we later propose a closed-form linear-regression calibrator, which yields more accurate calibration at a fraction of the cost.

### 3.3 PROPOSED METHOD

Training solely on new task data $\mathcal{D}_t$, as in AdaGauss (Rypeść et al., 2024b), can suffer from feature congestion between new and old classes, degrading the representation quality and overall performance. To address this limitation, we propose to apply projector-enhanced prototype-guided contrastive learning that applies contrastive learning on post-projection space, and an old-class repulsion loss that acts directly on the pre-projection space. Furthermore, we introduce a principled and efficient prototype calibration strategy, that incorporates similarity between prototypes and new task features to replace existing SGD-based calibration.

#### 3.3.1 PROJECTOR-ENHANCED CONTRASTIVE LEARNING

After training on task $t-1$, we obtain the feature extractor $f_{\theta_{t-1}}$ and Gaussian prototypes $(\boldsymbol{\mu}_c, \boldsymbol{\Sigma}_c)$ for each old class $c$. Prototype-guided contrastive learning, as in FCS (Li et al., 2024), enforces separation using new samples and old prototypes. However, applying both contrastive loss and the classification loss on the same feature space can suppress features important for classification (Xue et al., 2024; Ouyang et al., 2025), thereby hindering classification performance (see Section 4.3). This issue is also reflected in sub-optimal results of FCS (Li et al., 2024) as observed in Table 1. To overcome this, we decouple the two objectives through a projection head $\mathcal{P}_{\phi_t}$; classification is performed on the pre-projection space while contrastive loss operates on the post-projection space $\mathbf{z}_{t,i} = \mathcal{P}_{\phi_t}(f_{\theta_t}(\mathbf{x}_i))$ simultaneously.

This decoupling channels the strong augmentation-invariant pressure of the contrastive loss into a separate post-projection space, while benefiting model adaptation from rich, class-discriminative features in the pre-projection space. To further promote stability across tasks, we encode the augmented views through the old feature extractor $f_{\theta_{t-1}}$ rather than the evolving $f_{\theta_t}$, encouraging the evolving features to remain closer to earlier features.

Given a mini-batch $\mathcal{B} = \{(\mathbf{x}_i, \mathbf{x}_i^{(q)}, y_i)_{i=1}^B\}$, where $\mathbf{x}_i^{(q)}$ is an augmented view of input $\mathbf{x}_i$, we formulate the supervised contrastive loss $\mathcal{L}_{\text{Con}}$ as:

$$\mathcal{L}_{\text{Con}} = -\frac{1}{B}\sum_{i=1}^B \frac{1}{|P_i|}\sum_{p\in P_i}\log\left(\frac{\exp\left(\widehat{\mathbf{z}}_{t,i}^\top \widehat{\mathbf{z}}_{t-1,p}^{(q)}/\tau\right)}{\exp\left(\widehat{\mathbf{z}}_{t,i}^\top \widehat{\mathbf{z}}_{t-1,p}^{(q)}/\tau\right) + \sum_{\mathbf{v}\in\mathcal{V}_i}\exp\left(\widehat{\mathbf{z}}_{t,i}^\top \mathbf{v}/\tau\right)}\right) \quad (4)$$

where, $\widehat{\mathbf{z}}_{t,i} = \frac{\mathbf{z}_{t,i}}{\|\mathbf{z}_{t,i}\|_2}$ represents normalized projected features and $\tau$ represents temperature. $P_i = \{j : y_j = y_i\}$ represents the set of same class indices for input index $i$ and $\mathcal{V}_i = \{\mathbf{z}_{t-1,p}^{(q)} : y_j \neq y_i\} \cup \{\widehat{\mathbf{p}}_c : c \in \mathcal{C}_{1:t-1}\}$ represents the negative samples for index $i$, including different class features and old class prototype samples. For each old class $c$, we sample $\mathbf{s}_c \sim \mathcal{N}(\boldsymbol{\mu}_c, \boldsymbol{\Sigma}_c)$, and pass through the projector $\mathbf{p}_c = \mathcal{P}_{\phi_t}(\mathbf{s}_c)$ and normalize to get $\widehat{\mathbf{p}}_c$. Including the prototype samples as negatives, the model benefits from richer inter-task discriminative features.

#### 3.3.2 PRE-PROJECTION OLD CLASS REPULSION

While the projection-based contrastive learning enhances class separation, it may not sufficiently reduce inter-task overlap in the pre-projection space. We therefore add a direct inter-task repulsion term on the pre-projection space. This strategy repels new-task features from old-class distributions, but does not enforce attraction within the same class, thereby avoiding over-compression.

$$\mathcal{L}_{\text{OldRep}} = \frac{1}{B}\sum_{i=1}^B \log\left(\sum_{c\in\mathcal{C}_{1:t-1}}\exp(-d_c(\mathbf{x}_i))\right) \quad (5)$$

where, $d_c(\mathbf{x}) = \left(\widehat{f}_{\theta_t}(\mathbf{x}) - \widehat{\boldsymbol{\mu}}_c\right)^\top \widehat{\boldsymbol{\Sigma}}_c^{-1}\left(\widehat{f}_{\theta_t}(\mathbf{x}) - \widehat{\boldsymbol{\mu}}_c\right)$ is the Mahalanobis distance between normalized new class feature and normalized old class mean $\widehat{\boldsymbol{\mu}}_c$. Here, $\widehat{\boldsymbol{\Sigma}}_c$ represents class $c$'s normalized covariance as defined in (Goswami et al., 2023). This loss yields a soft, distance-weighted repulsion, adaptively emphasizing nearby old classes. The final loss gradient is the weighted average of gradients from each class $\nabla\mathcal{L}_{\text{OldRep}}(\mathbf{x}) = -\sum_{c=1}^{|\mathcal{C}_{1:t-1}|}\pi_c\nabla d_c(\mathbf{x})$ with weight

---

**Algorithm 1** Proposed Method

---

**Require:** Data stream $\mathcal{D} = \{\mathcal{D}_1, \ldots, \mathcal{D}_T\}$, Model $\{f_\theta, g_\varphi\}$, Hyper-parameters $\alpha, \beta, \gamma, \tau$
1: **for** each minibatch $\mathcal{B} \subset \mathcal{D}_1$ **do**
2:     Train with loss $\mathcal{L}_1 = \mathcal{L}_{\mathrm{Cls}} + \alpha \mathcal{L}_{\mathrm{Con}}$
3: **end for**
4: Create new prototypes $\{(\boldsymbol{\mu}_c, \boldsymbol{\Sigma}_c) \big| c \in \mathcal{C}_1\}$.
5: **for** $t = 2, \ldots, T$ **do**
6:     **for** each minibatch $\mathcal{B} \subset \mathcal{D}_t$ **do**
7:         Train with loss $\mathcal{L}_t = \mathcal{L}_{\mathrm{Cls}} + \alpha \mathcal{L}_{\mathrm{Con}} + \beta \mathcal{L}_{\mathrm{OldRep}} + \gamma \mathcal{L}_{\mathrm{KD}}$.
8:     **end for**
9:     Create new prototypes $\{(\boldsymbol{\mu}_c, \boldsymbol{\Sigma}_c) \big| c \in \mathcal{C}_t\}$.
10:     **for** $c \in \mathcal{C}_{1:t-1}$ **do**
11:         Get least-squares prototype calibrator $\widehat{\mathcal{W}_c}$ from Equation (8).
12:         Update $\boldsymbol{\mu}_c \leftarrow \widehat{\mathcal{W}_c}^\top \boldsymbol{\mu}_c, \boldsymbol{\Sigma}_c \leftarrow \widehat{\mathcal{W}_c}^\top \boldsymbol{\Sigma}_c \widehat{\mathcal{W}_c}$.
13:     **end for**
14: **end for**

---

$\pi_c = \frac{\exp(-d_c(\mathbf{x}))}{\sum_{k=1}^{|\mathcal{C}_{1:t-1}|} \exp(-d_k(\mathbf{x}))}$. The old classes close to the features exert larger gradient weight, while old classes away from the features have relatively less influence.

### 3.3.3 SIMILARITY-WEIGHTED LINEAR REGRESSION-BASED PROTOTYPE CALIBRATION

To efficiently align outdated class prototypes with evolving feature extractor, we introduce a similarity-weighted linear-regression-based prototype calibrator (SLrPC). Unlike prior methods, our method provides a closed-form solution, while weighting each new sample by its proximity to the old class distribution. For each class $c$, we adopt a linear prototype calibrator $\mathcal{T}'$ with parameters $\mathcal{W} \in \mathbb{R}^{M \times M}$ to project old prototypes to the new feature space. The weight $\omega_i^{(c)}$ assigned to a sample $\mathbf{x}_i$ reflects its likelihood under the old prototype distribution $\mathcal{N}(\boldsymbol{\mu}_c, \boldsymbol{\Sigma}_c)$:

$$\omega_i^{(c)} = \frac{\mathcal{N}(f_{\theta_{t-1}}(\mathbf{x}_i); \boldsymbol{\mu}_c, \boldsymbol{\Sigma}_c)}{\sum_{j=1}^{N_t} \mathcal{N}(f_{\theta_{t-1}}(\mathbf{x}_j); \boldsymbol{\mu}_c, \boldsymbol{\Sigma}_c)} \tag{6}$$

Stacking these weights together yields $\boldsymbol{\Omega}_c = \mathrm{diag}\left(\omega_i^{(c)}, \ldots, \omega_{N_t}^{(c)}\right) \in \mathbb{R}^{N_t \times N_t}$. Let $\mathcal{F}_{t-1} \in \mathbb{R}^{N_t \times M}$ and $\mathcal{F}_t \in \mathbb{R}^{N_t \times M}$ represent the feature matrices from old and new feature extractors, respectively. We define the similarity-weighted prototype calibration loss as:

$$\mathcal{L}_{\mathrm{SLrPC}} = \sum_{i=1}^{N_t} \omega_i^{(c)} \| f_{\theta_t}(\mathbf{x}_i) - \mathcal{W}^\top f_{\theta_{t-1}}(\mathbf{x}_i) \|^2 = \mathrm{tr}\left[ (\mathcal{F}_t - \mathcal{F}_{t-1}\mathcal{W})^\top \boldsymbol{\Omega}_c (\mathcal{F}_t - \mathcal{F}_{t-1}\mathcal{W}) \right]. \tag{7}$$

We use closed-form least-squares solution to get the prototype calibrator weight for each class $c$ as:

$$\widehat{\mathcal{W}_c} = \arg\min_{\mathcal{W}} \mathcal{L}_{\mathrm{SLrPC}} = \left(\mathcal{F}_{t-1}^\top \boldsymbol{\Omega}_c \mathcal{F}_{t-1}\right)^{-1} \mathcal{F}_{t-1}^\top \boldsymbol{\Omega}_c \mathcal{F}_t \tag{8}$$

Then, the class prototypes are updated as $\boldsymbol{\mu}_c' = \widehat{\mathcal{W}_c}^\top \boldsymbol{\mu}_c, \boldsymbol{\Sigma}_c' = \widehat{\mathcal{W}_c}^\top \boldsymbol{\Sigma}_c \widehat{\mathcal{W}_c}, \forall c \in \mathcal{C}_{1:t-1}$. A simple class-agnostic solution is obtained using an unweighted linear-regression solution with $\boldsymbol{\Omega} = \mathbf{I}_{N_t}$, where $\mathbf{I}_{N_t}$ is an identity matrix. Coupled with projector-aided contrastive learning and old class repulsion, our similarity-weighted prototype calibrator results in superior performance, rendering existing calibrators time-consuming. Algorithm 1 details the training procedure for our method.

## 4 EXPERIMENT

We conduct comprehensive experiments of our proposed method on three benchmark datasets: CIFAR100 (Krizhevsky, 2009), TinyImageNet (Le & Yang, 2015), and ImageNet100 (Deng et al., 2009). Our evaluation follows $T = 10$ and $T = 20$ task settings using *offline cold-start* EFCIL protocol. Next, we carry out an ablation study to analyze the effect of projector-enhanced contrastive learning, old-class repulsion, and similarity-weighted linear regression-based prototype calibration.

### 4.1 EXPERIMENTAL SETTING

**Implementation details**   We use ResNet-18 (He et al., 2016) as the feature extractor and set the feature dimension $M = 128$ via a single linear layer. The projection head for contrastive learning comprises a linear expansion layer (factor of $\rho = 32$), a GeLU activation (Hendrycks & Gimpel, 2016), and a final layer that reduces the output back to $M$. We set the contrastive-training loss weight $\alpha = 1$ and temperature $\tau = 0.1$. We use old class repulsion loss weight $\beta = 1$ for CIFAR100 and TinyImageNet and $\beta = 0.5$ for ImageNet100 dataset. We set knowledge distillation weight $\gamma = 10$ for CIFAR100 variants and $\gamma = 15$ for larger datasets TinyImageNet and ImageNet100. We train for 200 epochs using SGD with a weight decay of $5e - 4$ and batch size of 256. The learning rate starts at 0.1 and decays by a factor of 10 after 60, 120, and 180 epochs. We provide additional details in Appendix A. For evaluation, we use two standard metrics: final accuracy $\text{Acc}_T$ is calculated on the union of all classes $\mathcal{C}_{1:T}$ after training on the final task $T$ and, average incremental accuracy $\overline{\text{Acc}}_{1:T}$ is the average of the accuracies after training on each task.

Table 1: Comparison of final accuracy ($\text{Acc}_T$) and average incremental accuracy ($\overline{\text{Acc}}_{1:T}$) on CI-FAR100, TinyImageNet, and ImageNet100 (means over 5 class orders; standard deviations in Appendix B). Best in **bold**, second-best underlined.

| Methods | CIFAR100 | | | | TinyImageNet | | | | ImageNet100 | | | |
|---|---|---|---|---|---|---|---|---|---|---|---|---|
| | $\text{Acc}_T$ | | $\overline{\text{Acc}}_{1:T}$ | | $\text{Acc}_T$ | | $\overline{\text{Acc}}_{1:T}$ | | $\text{Acc}_T$ | | $\overline{\text{Acc}}_{1:T}$ | |
| | $T$=10 | $T$=20 | $T$=10 | $T$=20 | $T$=10 | $T$=20 | $T$=10 | $T$=20 | $T$=10 | $T$=20 | $T$=10 | $T$=20 |
| *Methods evolving feature extractor w/o prototype calibration* | | | | | | | | | | | | |
| EWC (Kirkpatrick et al., 2016) | 31.17 | 17.37 | 49.14 | 31.02 | 17.60 | 11.30 | 32.60 | 26.80 | 24.59 | 12.78 | 39.40 | 26.95 |
| LwF (Li & Hoiem, 2017) | 32.80 | 17.44 | 53.91 | 38.39 | 26.09 | 15.02 | 45.14 | 32.94 | 37.71 | 18.64 | 56.41 | 40.23 |
| PASS (Zhu et al., 2021b) | 30.45 | 17.44 | 47.86 | 32.86 | 24.11 | 18.73 | 39.25 | 32.01 | 26.40 | 14.38 | 45.74 | 31.65 |
| IL2A (Zhu et al., 2021a) | 31.70 | 23.00 | 48.40 | 40.20 | 25.30 | 19.80 | 42.00 | 35.50 | 27.70 | 17.50 | 48.40 | 34.90 |
| SSRE (Zhu et al., 2022) | 30.40 | 17.52 | 47.26 | 32.45 | 22.93 | 17.34 | 38.82 | 30.62 | 25.42 | 16.25 | 43.76 | 31.15 |
| NAPA (Malepathirana et al., 2023) | 37.09 | 22.25 | 51.69 | 39.53 | 25.39 | 18.11 | 36.77 | 30.36 | 27.30 | 19.96 | 43.30 | 34.23 |
| PRAKA (Shi & Ye, 2023) | 41.58 | 32.37 | 54.22 | 45.38 | 30.85 | 24.69 | 43.76 | 36.44 | 38.90 | 30.72 | 53.46 | 45.99 |
| PRL (Shi & Ye, 2024) | 43.43 | 32.58 | 55.85 | 44.58 | 36.12 | 29.79 | 47.29 | 41.72 | 48.21 | 38.16 | 60.91 | 51.76 |
| *Methods with fixed feature extractor* | | | | | | | | | | | | |
| FeTrIL (Petit et al., 2023) | 34.94 | 23.28 | 51.20 | 38.48 | 30.97 | 25.70 | 45.60 | 39.54 | 36.17 | 26.63 | 52.63 | 42.43 |
| FeCAM (Goswami et al., 2023) | 32.40 | 20.60 | 48.30 | 34.10 | 30.80 | 25.20 | 44.50 | 38.30 | 38.70 | 29.00 | 54.80 | 44.60 |
| DS-AL (Zhuang et al., 2024) | 40.80 | 31.70 | 54.90 | 43.20 | 33.60 | 26.50 | 47.20 | 41.60 | 46.80 | 36.70 | 58.60 | 48.50 |
| *Methods evolving feature extractor with prototype calibration* | | | | | | | | | | | | |
| SDC (Yu et al., 2020) | 42.36 | 33.88 | 57.53 | 48.47 | 29.85 | 15.36 | 44.28 | 26.95 | 43.50 | 19.90 | 61.37 | 39.48 |
| FCS (Li et al., 2024) | 36.15 | 20.22 | 51.22 | 33.47 | 26.54 | 15.41 | 38.02 | 26.57 | 35.74 | 22.77 | 52.00 | 40.07 |
| EFC (Magistri et al., 2024) | 44.07 | 33.93 | 59.25 | 49.93 | 33.33 | 28.88 | 46.87 | 41.72 | 50.16 | 39.14 | 63.64 | 52.68 |
| AdaGauss (Rypeść et al., 2024b) | 51.50 | 42.40 | 63.40 | 55.08 | 39.84 | 31.72 | 52.25 | 45.51 | 50.78 | 41.53 | 65.21 | 57.15 |
| Ours | **56.58** | **46.87** | **67.54** | **59.39** | **41.46** | **34.64** | **53.71** | **47.72** | **53.30** | **44.13** | **67.86** | **60.09** |

### 4.2 COMPARISON WITH SOTA

We compare against three groups of baselines: (i) methods that evolve feature extractor without prototype calibration (e.g. EWC (Kirkpatrick et al., 2016), PASS (Zhu et al., 2021b), PRAKA (Shi & Ye, 2023), PRL (Shi & Ye, 2024) etc.), (ii) methods that freeze the feature extractor after training on the first task (FeTrIL (Petit et al., 2023), FeCAM (Goswami et al., 2023), DS-AL (Zhuang et al., 2024)) and (iii) methods that evolve feature extractor along with prototype calibration: (SDC (Yu et al., 2020), EFC (Magistri et al., 2024), FCS (Li et al., 2024), AdaGauss (Rypeść et al., 2024b)).

Table 1 shows that our method outperforms all existing approaches in both average incremental accuracy and final accuracy. In particular, we significantly surpass methods that lack prototype calibration (e.g., PASS, PRAKA, PRL) and those that freeze the feature extractor after the first task (e.g., FeTrIL, FeCAM, DS-AL), which results in insufficiently generalizable features. Against recent prototype-calibration techniques such as EFC, FCS, and AdaGauss, our method still achieves better performance. Specifically, on the 10-task splits of CIFAR100, TinyImageNet, and ImageNet100 datasets, we record improvements in average incremental accuracy of 4.14 %, 1.46 %, and 2.65 %, respectively, over the previous SOTA. These gains stem from enhanced class feature representations from our projector-aided contrastive-learning and old-class repulsion loss, as well as more effective prototype calibration via our similarity-weighted, linear regression-based calibrator (see Section 4.3). Figure 2 further illustrates that our proposed method consistently outperforms compet-

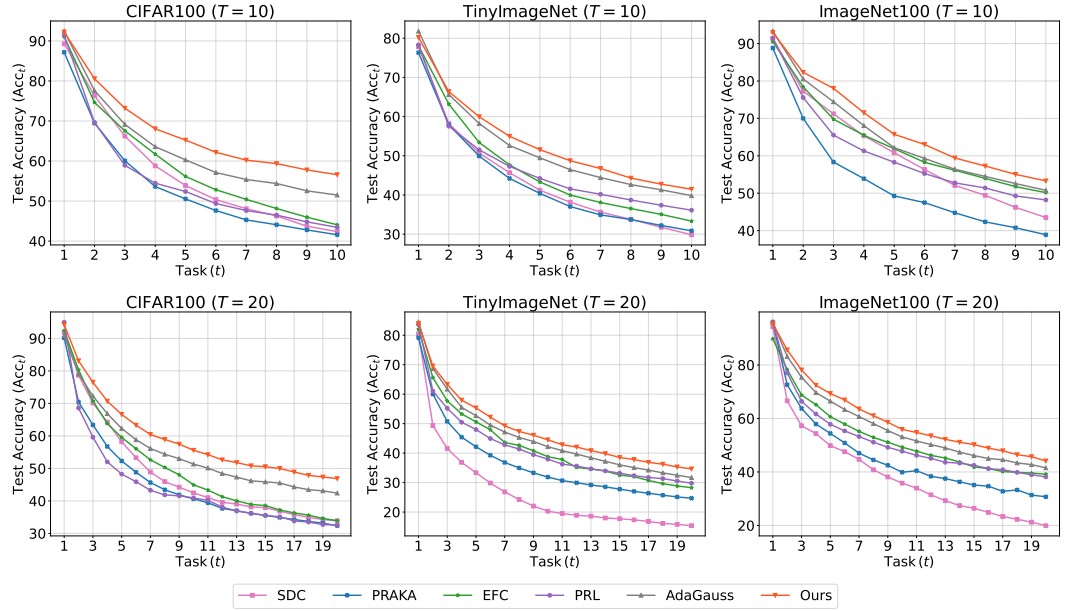

Figure 2: Detailed comparison of average test accuracies ($\mathrm{Acc}_t$) with representative methods that update the feature-extractor across tasks on CIFAR100, TinyImageNet, and ImageNet100 datasets.

ing approaches across the EFCIL tasks, achieving higher test accuracies than representative methods that evolve the feature extractor, both with and without prototype calibration.

### 4.3 ABLATION STUDY

To further analyze each component's impact, we investigate the performance on CIFAR100 and TinyImageNet datasets under different settings. Table 4.3 shows that using our similarity-weighted linear regression-based calibrator improves performance compared to AdaGauss's iterative calibrator and also compared to simple unweighted linear regression-based calibration. Furthermore, applying contrastive learning and classification losses on the same feature space degrades the performance compared to the case without contrastive learning. In contrast, pairing contrastive learning through a projection head with our similarity-weighted linear regression-based prototype calibrator improves the performance further. Finally, our old-class repulsion loss combined with projection-head enhanced contrastive training and our calibration method achieves the best overall performance.

Table 2: Ablation Study: Comparison of EFCIL performances on CIFAR100 and TinyImageNet using AdaGauss calibrator, and variants of our linear regression-based calibrators, and with and without contrastive learning, projection head, and old-class repulsion.

| Settings | | | | CIFAR100 ($T = 20$) | | TinyImageNet ($T = 20$) | |
|---|---|---|---|---|---|---|---|
| Prototype Calibrator | Contrast. Learning | Project. Head | old-class Repulsion | $\mathrm{Acc}_T$ | $\overline{\mathrm{Acc}}_{1:T}$ | $\mathrm{Acc}_T$ | $\overline{\mathrm{Acc}}_{1:T}$ |
| AdaGauss (baseline) | | | | $42.40 \pm 0.84$ | $55.08 \pm 2.54$ | $31.72 \pm 0.57$ | $45.51 \pm 0.37$ |
| | ✗ | ✗ | ✗ | $43.81 \pm 0.77$ | $56.17 \pm 1.78$ | $33.90 \pm 0.74$ | $45.69 \pm 0.37$ |
| Sim-weighted | ✓ | ✗ | ✗ | $43.44 \pm 1.06$ | $55.78 \pm 2.19$ | $33.23 \pm 0.63$ | $44.29 \pm 0.64$ |
| Linear Reg. | ✓ | ✓ | ✗ | $44.65 \pm 0.87$ | $57.89 \pm 2.03$ | $34.35 \pm 0.66$ | $47.16 \pm 0.69$ |
| | ✓ | ✓ | ✓ | $\mathbf{46.87 \pm 1.37}$ | $\mathbf{59.39 \pm 1.99}$ | $\mathbf{34.64 \pm 0.71}$ | $\mathbf{47.72 \pm 0.85}$ |
| Linear Reg. | ✓ | ✓ | ✓ | $45.10 \pm 1.10$ | $57.87 \pm 2.03$ | $33.60 \pm 1.03$ | $47.07 \pm 0.92$ |

**Effectiveness of linear regression-based calibration**   Figure 3 compares the prototype calibration quality between AdaGauss, our unweighted, and similarity-weighted linear regression-based calibrators. Although the similarity-weighted and unweighted linear regression variants perform similarly, the similarity-weighted version further narrows the gap between the calibrated and true Gaussian distributions of the old-classes—far more than the original AdaGauss calibrator. The ef-

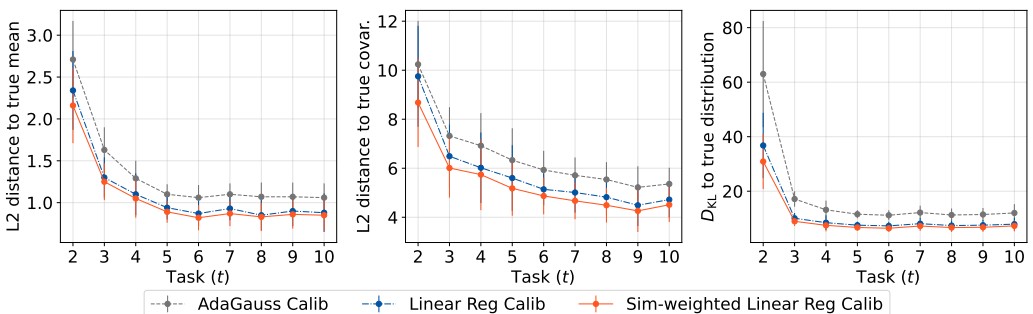

Figure 3: Calibration quality on TinyImageNet ($T$=10). We compare L2 distance for class means (*left*) and covariances (*middle*), and KL-divergence $D_{\text{KL}}$ (*right*) between true class distribution and calibration results of different methods, trained using similarity-weighted linear regression-based calibration. Lower is better.

fect of prototype calibration is reflected in Table 2, which shows consistent gains in both final accuracy and average incremental accuracy using our linear regression-based calibrator. This concludes that our linear regression calibrator is not only sufficient but superior for prototype calibration. We further report $57\times$ faster efficiency than AdaGauss on TinyImageNet ($T = 10$) and provide a time complexity comparison of our prototype calibrator in Appendix D.

**Impact of projection-enhanced contrastive training** Decoupling contrastive learning from classification via a projection head improves accuracy (Table 2). This aligns with the intuition that the projection head allows the benefits of contrastive learning by preserving a richer class-discriminative structure in the pre-projection space (Ouyang et al., 2025). Appendix F analyzes the impact of architecture for the projection head, and Appendix E shows further gains from the asymmetric embedding strategy in contrastive learning, encoding augmented views with $f_{\theta_{t-1}}$ instead of $f_{\theta_t}$ in Equation 4.

**Impact of old-class Repulsion Loss** Adding old-class repulsion shows improvement in accuracy. To evaluate its effect on class geometry, we further evaluate the separation of class features. Specifically, we calculate the cosine distance between true class means after final training. Figure 4 illustrates that the inter-class-mean distance increases significantly across the classes after including old-class repulsion loss. This increase in inter-class means' angular distance illustrates that the old-class repulsion loss increases the inter-class separation across old and new classes in the feature space.

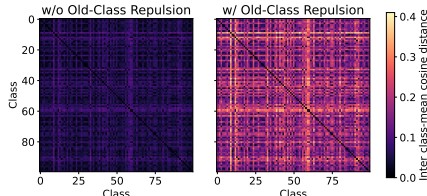

Figure 4: Impact of old-class repulsion loss on inter-class cosine distance on CIFAR100 ($T = 10$) dataset.

## 5 CONCLUSION AND LIMITATIONS

We integrate three strategies to improve adaptation and tackle feature-space congestion and prototype drift. First, we introduce projector-enhanced contrastive learning: we jointly train contrastive learning and classification losses on two distinct feature spaces, separated by a projection head. This allows learning richer class-discriminative embeddings in the pre-projection space. Second, we introduce the old class repulsion loss that explicitly mitigates congestion between new and old class features. Third, we replace the iterative stochastic calibration with similarity-weighted closed-form regression calibrator, which calibrates old-class prototypes more accurately and over an order of magnitude faster. Together, these components form a coherent EFCIL approach that outperforms existing SOTA methods across several standard benchmarks.

**Limitations** Our work is primarily empirical and does not provide a formal theoretical analysis, which we leave for future investigation. In addition, incorporating the contrastive learning objective introduces extra computational overhead compared to purely classification-based approaches.

## 6 ETHICS STATEMENT

This research adheres to the ICLR code of ethics and does not harm any societal groups while acknowledging the necessity to use research in the benefit of the society and its members. We aim to uphold high standards of scientific excellence by attaining transparency and reproducibility, without harming the natural environment in any way. We highly value equality, tolerance, and justice and respect the privacy of each individual. We have been fair in terms of acknowledgment and authorship. By aligning with these principles, we aim to foster quality research work while pursuing socially responsible work aimed at benefiting the humanity.

## 7 REPRODUCIBILITY STATEMENT

We provide the details of the hyperparameters in Section 4.1. Additional implementation details are included in Appendix A, along with a link to the codebase to support reproducibility in the abstract.

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

# A    ADDITIONAL DETAILS

**Datasets**    We evaluate our proposed method on three benchmark datasets: CIFAR100 (Krizhevsky, 2009), TinyImageNet (Le & Yang, 2015) and ImageNet100 (Deng et al., 2009). CIFAR100 consists of $32 \times 32$-pixel images divided into 100 classes, each containing 500 training images and 100 test images. TinyImageNet comprises $64 \times 64$-pixel images divided into 200 classes, each containing 500 training images and 50 test images. ImageNet100 contains 100 classes randomly sampled from ImageNet (seed=1993) with 1300 training images and 50 test images per class. All ImageNet100 images are cropped to $224 \times 224$ pixels.

**Protocol**    We follow the *offline cold-start* EFCIL protocol used in (Magistri et al., 2024; Rypeść et al., 2024b) to evaluate our method against the existing baseline methods; we split the full set of classes equally across the tasks, which offers a more realistic evaluation for evolving feature-extractor methods, since they no longer rely on seeing half the classes in the first task to learn generalizable features. We experiment with 10 tasks ($T = 10$) and 20 tasks ($T = 20$). For CIFAR100 and ImageNet100, each task contains 20 classes when $T = 10$ and and 10 classes when $T = 20$. For TinyImageNet, each task contains 20 in $T = 10$ split and 10 classes in $T = 20$ setting.

**Evaluation Metrics**    We evaluate the performances of our method against the other methods using two standard metrics: final accuracy $\text{Acc}_T$ and average incremental accuracy $\overline{\text{Acc}}_{1:T}$. The final accuracy $\text{Acc}_T$ is calculated on the union of all classes $\mathcal{C}_{1:T}$ after training on the final task $T$. And, average incremental accuracy $\overline{\text{Acc}}_{1:T}$ summarizes the course of incremental training by averaging the accuracies after training on each task.

$$\overline{\text{Acc}}_{1:T} = \frac{1}{T} \sum_{t=1}^{T} \text{Acc}_t, \tag{9}$$

where, $\text{Acc}_t$ is the accuracy over all observed classes up to task $t$ i.e., $\mathcal{C}_{1:t}$.

**Remaining Implementation Details**    To mitigate the bias arising from the choice of class order, we ran all experiments with five different class orderings, obtained by random permutations with seeds 1993, 42, 7, 8, and 10. We consistently use these seeds across all datasets and task splits for the methods we trained and evaluated on. Furthermore, we apply self-supervised rotations (Lee et al., 2020) only during the first task (Magistri et al., 2024; Rypeść et al., 2024b). For contrastive learning, we use stronger augmentations like random resized crop, random grayscale, and color jitter.

# B    DETAILED RESULTS

The results for CIFAR100 are provided in Table 3, TinyImageNet in Table 4, and ImageNet100 in 5. For some results other than DS-AL (Zhuang et al., 2024), where we report a single number, the result was calculated using a single run, as the methods took a long time to run. We report both final accuracy and average incremental accuracy as evaluation metrics.

For the older baselines—EWC, LwF, PASS, IL2A, SSRE, FeTrIL, FeCAM, and DS-AL—we quote the results in the EFC and AdaGauss papers. However, we reran EFC and AdaGauss (with feature dimension $M = 128$) and all remaining methods with the same class ordering using their available code-bases, as EFC and AdaGauss are the most recent methods. For FCS, we found that it performs better without self-supervised label augmentation in the incremental tasks, so we report the stronger variant. Finally, consistent with the authors' observations, our own experiments showed that PRL performs better without the prototype calibration step, so we categorize it accordingly.

Table 3: Comparison of final accuracy ($\text{Acc}_T$) and average incremental accuracy ($\overline{\text{Acc}}_{1:T}$) on CIFAR100 (means and standard deviations over 5 class orders). Best results are highlighted in **bold**.

| Methods | Final Accuracy ($\text{Acc}_T$) | | Avg Inc Accuracy ($\overline{\text{Acc}}_{1:T}$) | |
|---|---|---|---|---|
| | $T = 10$ | $T = 20$ | $T = 10$ | $T = 20$ |
| *Methods evolving feature extractor w/o prototype calibration* | | | | |
| EWC (Kirkpatrick et al., 2016) | $31.17 \pm 2.94$ | $17.37 \pm 2.43$ | $49.14 \pm 1.28$ | $31.02 \pm 1.15$ |
| LwF (Li & Hoiem, 2017) | $32.80 \pm 3.08$ | $17.44 \pm 0.73$ | $53.91 \pm 1.67$ | $38.39 \pm 1.05$ |
| PASS (Zhu et al., 2021b) | $30.45 \pm 1.01$ | $17.44 \pm 0.69$ | $47.86 \pm 1.93$ | $32.86 \pm 1.03$ |
| IL2A (Zhu et al., 2021a) | $31.70 \pm 1.30$ | $23.00 \pm 0.90$ | $48.40 \pm 2.00$ | $40.20 \pm 1.10$ |
| SSRE (Zhu et al., 2022) | $30.40 \pm 0.74$ | $17.52 \pm 0.80$ | $47.26 \pm 1.91$ | $32.45 \pm 1.07$ |
| NAPA (Malepathirana et al., 2023) | $37.09$ | $22.25$ | $51.69$ | $39.53$ |
| PRAKA (Shi & Ye, 2023) | $41.58 \pm 1.80$ | $32.37 \pm 1.83$ | $54.22 \pm 2.46$ | $45.38 \pm 2.74$ |
| PRL (Shi & Ye, 2024) | $43.43 \pm 2.70$ | $32.58 \pm 3.61$ | $55.85 \pm 2.39$ | $44.58 \pm 4.20$ |
| *Methods with fixed feature extractor* | | | | |
| FeTrIL (Petit et al., 2023) | $34.94 \pm 0.46$ | $23.28 \pm 1.24$ | $51.20 \pm 1.13$ | $38.48 \pm 1.07$ |
| FeCAM (Goswami et al., 2023) | $32.40 \pm 0.40$ | $20.60 \pm 0.70$ | $48.30 \pm 0.90$ | $34.10 \pm 1.10$ |
| DS-AL (Zhuang et al., 2024) | $40.80$ | $31.70$ | $54.90$ | $43.20$ |
| *Methods evolving feature extractor with prototype calibration* | | | | |
| SDC (Yu et al., 2020) | $42.36 \pm 0.85$ | $33.88 \pm 1.04$ | $57.53 \pm 1.02$ | $48.47 \pm 1.23$ |
| FCS (Li et al., 2024) | $36.15 \pm 2.88$ | $20.22 \pm 4.94$ | $51.22 \pm 2.24$ | $33.47 \pm 7.04$ |
| EFC (Magistri et al., 2024) | $44.07 \pm 1.30$ | $33.93 \pm 1.42$ | $59.25 \pm 1.59$ | $49.93 \pm 1.93$ |
| AdaGauss (Rypeść et al., 2024b) | $51.50 \pm 1.00$ | $42.40 \pm 0.84$ | $63.50 \pm 1.04$ | $55.08 \pm 2.54$ |
| Ours | $\mathbf{56.58 \pm 0.66}$ | $\mathbf{46.87 \pm 1.37}$ | $\mathbf{67.54 \pm 0.93}$ | $\mathbf{59.39 \pm 1.99}$ |

Table 4: Comparison of final accuracy ($\text{Acc}_T$) and average incremental accuracy ($\overline{\text{Acc}}_{1:T}$) on TinyImageNet (means and standard deviations over 5 class orders). Best results are highlighted in **bold**.

| Methods | Final Accuracy ($\text{Acc}_T$) | | Avg Inc Accuracy ($\overline{\text{Acc}}_{1:T}$) | |
|---|---|---|---|---|
| | $T = 10$ | $T = 20$ | $T = 10$ | $T = 20$ |
| *Methods evolving feature extractor w/o prototype calibration* | | | | |
| EWC (Kirkpatrick et al., 2016) | $17.60 \pm 1.50$ | $11.30 \pm 1.20$ | $32.60 \pm 1.20$ | $26.80 \pm 1.10$ |
| LwF (Li & Hoiem, 2017) | $26.09 \pm 1.29$ | $15.02 \pm 0.67$ | $45.14 \pm 0.88$ | $32.94 \pm 0.54$ |
| PASS (Zhu et al., 2021b) | $24.11 \pm 0.48$ | $18.73 \pm 1.43$ | $39.25 \pm 0.90$ | $32.01 \pm 1.68$ |
| IL2A (Zhu et al., 2021a) | $25.30 \pm 0.90$ | $19.80 \pm 1.80$ | $42.00 \pm 1.70$ | $35.50 \pm 2.30$ |
| SSRE (Zhu et al., 2022) | $22.93 \pm 0.95$ | $17.34 \pm 1.06$ | $38.82 \pm 1.99$ | $30.62 \pm 1.96$ |
| NAPA (Malepathirana et al., 2023) | $25.39$ | $18.11$ | $36.77$ | $30.36$ |
| PRAKA (Shi & Ye, 2023) | $30.85 \pm 0.95$ | $24.69 \pm 0.86$ | $43.76 \pm 1.11$ | $36.44 \pm 1.18$ |
| PRL (Shi & Ye, 2024) | $36.12 \pm 0.62$ | $29.79 \pm 1.74$ | $47.29 \pm 0.57$ | $41.72 \pm 1.28$ |
| *Methods with fixed feature extraction* | | | | |
| FeTrIL (Petit et al., 2023) | $30.97 \pm 0.90$ | $25.70 \pm 0.61$ | $45.60 \pm 1.67$ | $39.54 \pm 1.19$ |
| FeCAM (Goswami et al., 2023) | $30.80 \pm 0.80$ | $25.20 \pm 0.60$ | $44.50 \pm 1.50$ | $38.30 \pm 1.10$ |
| DS-AL (Zhuang et al., 2024) | $33.60$ | $26.50$ | $47.20$ | $41.60$ |
| *Methods evolving feature extractor with prototype calibration* | | | | |
| SDC (Yu et al., 2020) | $29.85 \pm 0.35$ | $15.36 \pm 0.97$ | $44.28 \pm 1.50$ | $26.95 \pm 1.30$ |
| FCS (Li et al., 2024) | $26.54 \pm 0.82$ | $15.41 \pm 1.75$ | $38.02 \pm 0.99$ | $26.57 \pm 1.67$ |
| EFC (Magistri et al., 2024) | $33.33 \pm 1.08$ | $28.88 \pm 0.83$ | $46.87 \pm 0.30$ | $41.72 \pm 0.92$ |
| AdaGauss (Rypeść et al., 2024b) | $39.84 \pm 0.47$ | $31.72 \pm 0.57$ | $52.25 \pm 0.55$ | $45.51 \pm 0.37$ |
| Ours | $\mathbf{41.46 \pm 0.46}$ | $\mathbf{34.64 \pm 0.71}$ | $\mathbf{53.71 \pm 0.64}$ | $\mathbf{47.72 \pm 0.85}$ |

Table 5: Comparison of final accuracy ($\mathrm{Acc}_T$) and average incremental accuracy ($\overline{\mathrm{Acc}}_{1:T}$) on ImageNet100 (means and standard deviations over 5 class orders). Best results are highlighted in **bold**.

| Methods | Final Accuracy ($\mathrm{Acc}_T$) | | Avg Inc Accuracy ($\overline{\mathrm{Acc}}_{1:T}$) | |
|---|---|---|---|---|
| | $T = 10$ | $T = 20$ | $T = 10$ | $T = 20$ |
| *Methods evolving feature extractor w/o prototype calibration* | | | | |
| EWC (Kirkpatrick et al., 2016) | $24.59 \pm 4.13$ | $12.78 \pm 1.95$ | $39.40 \pm 3.05$ | $26.95 \pm 1.02$ |
| LwF (Li & Hoiem, 2017) | $37.71 \pm 2.53$ | $18.64 \pm 1.67$ | $56.41 \pm 1.03$ | $40.23 \pm 0.43$ |
| PASS (Zhu et al., 2021b) | $26.40 \pm 1.33$ | $14.38 \pm 1.22$ | $45.74 \pm 0.18$ | $31.65 \pm 0.42$ |
| IL2A (Zhu et al., 2021a) | $27.70 \pm 1.80$ | $17.50 \pm 1.60$ | $48.40 \pm 1.50$ | $34.90 \pm 0.70$ |
| SSRE (Zhu et al., 2022) | $25.42 \pm 1.17$ | $16.25 \pm 1.05$ | $43.76 \pm 1.07$ | $31.15 \pm 1.53$ |
| NAPA (Malepathirana et al., 2023) | $27.30$ | $19.96$ | $43.30$ | $34.23$ |
| PRAKA (Shi & Ye, 2023) | $38.90$ | $30.72$ | $53.46$ | $45.99$ |
| PRL (Shi & Ye, 2024) | $48.21 \pm 0.59$ | $38.16 \pm 2.57$ | $60.91 \pm 0.60$ | $51.76 \pm 2.30$ |
| *Methods with fixed feature extraction* | | | | |
| FeTrIL (Petit et al., 2023) | $36.17 \pm 1.18$ | $26.63 \pm 1.45$ | $52.63 \pm 0.56$ | $42.43 \pm 2.05$ |
| FeCAM (Goswami et al., 2023) | $38.70 \pm 1.00$ | $29.00 \pm 1.30$ | $54.80 \pm 0.50$ | $44.60 \pm 2.00$ |
| DS-AL (Zhuang et al., 2024) | $46.80$ | $36.70$ | $58.60$ | $48.50$ |
| *Methods evolving feature extractor with prototype calibration* | | | | |
| SDC (Yu et al., 2020) | $43.50 \pm 0.96$ | $19.90 \pm 1.42$ | $61.37 \pm 1.36$ | $39.48 \pm 2.17$ |
| FCS (Li et al., 2024) | $35.74 \pm 3.58$ | $22.77 \pm 1.92$ | $52.00 \pm 3.45$ | $40.07 \pm 2.51$ |
| EFC (Magistri et al., 2024) | $50.16 \pm 0.75$ | $39.14 \pm 0.71$ | $63.64 \pm 0.75$ | $52.68 \pm 1.78$ |
| AdaGauss (Rypeść et al., 2024b) | $50.78 \pm 0.84$ | $41.53 \pm 1.01$ | $65.21 \pm 0.42$ | $57.15 \pm 0.61$ |
| Ours | $\mathbf{53.30 \pm 1.02}$ | $\mathbf{44.13 \pm 0.81}$ | $\mathbf{67.86 \pm 0.72}$ | $\mathbf{60.09 \pm 0.87}$ |

## C GENERALIZED VIEWPOINTS FOR PROTOTYPE CALIBRATION

### C.1 GENERALIZED SIMILARITY-WEIGHTED DRIFT ESTIMATION

Assuming that the feature vector, for each past task class $c$, is normally distributed $\mathcal{N}(\cdot; \boldsymbol{\mu}_c, \boldsymbol{\Sigma}_c)$, we provide the general framework for similarity-weighted prototype update.

$$\boldsymbol{\mu}_c^{'} = \boldsymbol{\mu}_c + \nu \sum_{i=1}^{N_t} \omega_i^{(c)} \delta_i^{t-1 \to t}, \qquad \delta_i^{t-1 \to t} = f_{\theta_t}(\mathbf{x}_i) - f_{\theta_{t-1}}(\mathbf{x}_i)$$

where $\nu$ is a hyper-parameter to scale the estimated drift. We then define the normalized weight $\omega_i^{(c)}$ for each sample $\mathbf{x}_i$, based on the similarity between the feature embedding from the old-task feature extractor $f_{t-1}$ and the saved prototype for class $c$ as:

$$\omega_i^{(c)} = \frac{\mathcal{N}(f_{\theta_{t-1}}(\mathbf{x}_i); \boldsymbol{\mu}_c, \boldsymbol{\Sigma}_c)}{\sum_{j=1}^{N_t} \mathcal{N}(f_{\theta_{t-1}}(\mathbf{x}_j); \boldsymbol{\mu}_c, \boldsymbol{\Sigma}_c)} = \frac{\exp\left(-\frac{1}{2}\left(f_{\theta_{t-1}}(\mathbf{x}_i) - \boldsymbol{\mu}_c\right)^\top \boldsymbol{\Sigma}_c^{-1}\left(f_{\theta_{t-1}}(\mathbf{x}_i) - \boldsymbol{\mu}_c\right)\right)}{\sum_{j=1}^{N_t} \exp\left(-\frac{1}{2}\left(f_{\theta_{t-1}}(\mathbf{x}_j) - \boldsymbol{\mu}_c\right)^\top \boldsymbol{\Sigma}_c^{-1}\left(f_{\theta_{t-1}}(\mathbf{x}_j) - \boldsymbol{\mu}_c\right)\right)}$$
(10)

Note that the determinant terms out of the exponential cancel out. Using $\Delta_{i,c} = f_{\theta_{t-1}}(\mathbf{x}_i) - \boldsymbol{\mu}_c$ to indicate the gap between current task sample and stored prototype in the old feature space, we can reformulate $\omega_i^{(c)}$ as:

$$\omega_i^{(c)} = \frac{\exp\left(-\frac{1}{2}\Delta_{i,c}^\top \boldsymbol{\Sigma}_c^{-1} \Delta_{i,c}\right)}{\sum_{j=1}^{N_t} \exp\left(-\frac{1}{2}\Delta_{j,c}^\top \boldsymbol{\Sigma}_c^{-1} \Delta_{j,c}\right)}$$
(11)

**Adaptation to existing methods:**

1. SDC Yu et al. (2020) assumes the feature space to have isotropic covariance $\boldsymbol{\Sigma}_c = \sigma \mathbf{I}$.

2. EFC Magistri et al. (2024) is an instantiation of this general framework where Elastic Feature Matrix (FIM, an alternative for features) is used in place of the inverse-covariance

matrix. Additionally, it does not apply class-specific covariance matrices $\boldsymbol{\Sigma}_c$ but rather a shared cumulative EFM $\mathbf{E}_{t-1}$, i.e.

$$\boldsymbol{\Sigma}_c^{-1} = \mathbf{E}_{t-1}$$

And, it sets $\nu = 1$ and introduces a new hyper-parameter $\sigma$ to mitigate the change in class-covariance because of using a shared EFM.

3. TEEN Wang et al. (2023) redefines $\Delta_{i,c} = \frac{f_{\theta_{t-1}}(\mathbf{x}_i)}{\|f_{\theta_{t-1}}(\mathbf{x}_i)\|} - \frac{\boldsymbol{\mu}_c}{\|\boldsymbol{\mu}_c\|}$ and assumes the features to have isotropic covariance such that $\boldsymbol{\Sigma}_c = \frac{1}{\sqrt{\tau}}\mathbf{I}$. Additionally, it calculates the drift in feature from task $t-1$ to $t$ $\delta_i^{t-1 \to t}$ using the class mean $\boldsymbol{\mu}_c$ as $\delta_i^{t-1 \to t} = f_{\theta_t}(\mathbf{x}_i) - \boldsymbol{\mu}_c$.

## C.2 Generalized Similarity-weighted Learnable Prototype Calibration

Current learnable calibration methods (Gomez-Villa et al., 2024; Li et al., 2024; Shi & Ye, 2024; Rypeść et al., 2024b) train a sub-network $\mathcal{T}$ to map old-class prototypes to new-task feature space by minimizing the calibration loss.

$$\mathcal{L}_{\text{LPC}} = \mathbb{E}_{(\mathbf{x}_i,y) \sim \mathcal{D}_t} \|f_{\theta_t}(\mathbf{x}_i) - \mathcal{T} \circ f_{\theta_{t-1}}(\mathbf{x}_i)\|^2 \tag{12}$$

After the calibrator training is complete, the old class feature samples $\mathbf{s}$ are updated as $\mathbf{s}' = \mathcal{T}(\mathbf{s})$. Here, $\mathbf{s} = \boldsymbol{\mu}_c$ if only class means are stored and $\mathbf{s} \sim \mathcal{N}(\boldsymbol{\mu}_c, \boldsymbol{\Sigma}_c)$ if full Gaussian distributions are stored as class prototypes.

We define a more general similarity-weighted learnable prototype calibrator with the loss function for each class c as:

$$\mathcal{L}_{\text{SLPC}}^{(c)} = \mathbb{E}_{(\mathbf{x}_i,y) \sim \mathcal{D}_t} \left\| \omega_i^{(c)} \left( f_{\theta_t}(\mathbf{x}_i) - \mathcal{T} \circ f_{\theta_{t-1}}(\mathbf{x}_i) \right) \right\|^2 \tag{13}$$

where, $\omega_i^{(c)}$ is the similarity weight defined in Equation 10. Following this, we obtain the closed-formed similarity-weighted linear regression-based calibrator in Section 3.3.3.

# D  Space and Time Complexity

**Memory requirements**   Our approach just adds a single down-sampling layer to reduce the feature dimensionality to $M$ in the feature extractor. After training, we discard the prototype calibrator and the projectors for contrastive learning and knowledge distillation. Consequently, once EFCIL training is completed, inference requires storing the feature extractor, and $|\mathcal{C}_{1:T}|M$ parameters for class means and $|\mathcal{C}_{1:T}|\frac{M \times (M+1)}{2}$ parameters for inverse of class covariances like in FeCAM(Goswami et al., 2023) and AdaGauss (Rypeść et al., 2024b).

**Overall training time**   Table 6 shows that the overall training time for our method is longer compared to AdaGauss and EFC, but much shorter compared to PRAKA. On average, our method takes 11% more time to train compared to AdaGauss. We use an A100 GPU with 12 workers to train the models. Though the contrastive learning loss increases the training time, we mitigate it with faster similarity-weighted linear regression-based prototype calibration, which removes the computational overhead due to SGD updates for calibration.

Table 6: Comparison of total training times for different methods on a 10-split of TinyImageNet.

| Methods | PRAKA | PRL | EFC | AdaGauss | Ours |
|---|---|---|---|---|---|
| Training time (hours) | $9.68 \pm 0.36$ | $5.00 \pm 0.07$ | $3.52 \pm 0.15$ | $4.70 \pm 0.14$ | $5.25 \pm 0.24$ |

**Prototype calibration time complexity**   For similarity-weighted prototype calibration, the time complexity for calibrating stored Gaussians is $\mathcal{O}(|C_{1:t-1}|)$, which compared to AdaGauss calibrator is $\mathcal{O}(\mathcal{E} + |\mathcal{C}_{1:t-1}|R)$, where $\mathcal{E}$ is the number of epochs required to train the calibrator and $R$ is the number of samples required for Gaussian calibration which works by sampling and mapping and

re-estimating Gaussian. We compare the average wall clock times required for prototype calibration in Table 7, which demonstrates our proposed similarity-weighted linear regression-based calibrator is approximately $57\times$ faster than the AdaGauss calibrator while taking a marginally longer time compared to the unweighted linear-regression version.

Table 7: Comparison of times required by different methods for full Gaussian calibration across tasks on the 10-task split of TinyImageNet dataset.

| Strategies | Linear Reg | Sim-weighted Linear Reg | AdaGauss Calib. |
|---|---|---|---|
| Calibration Time (seconds) | $4.90 \pm 0.06$ | $5.52 \pm 0.78$ | $316.55 \pm 1.14$ |

## E  CHOICE OF FEATURE EXTRACTOR FOR AUGMENTED INPUTS

We compare two choices for the feature extractor used to compute the augmented-image features: (i) current, evolving feature extractor $f_{\theta_t}$ and (ii) old feature extractor $f_{\theta_{t-1}}$. Figure 5 shows that using the old feature extractor $f_{\theta_{t-1}}$ yields higher final accuracy than using the evolving feature extractor $f_{\theta_t}$. We hypothesize that encouraging $f_{\theta_t}$'s outputs to match that of $f_{\theta_{t-1}}$ implicitly reinforces the knowledge distillation signal and thus helps in knowledge retention.

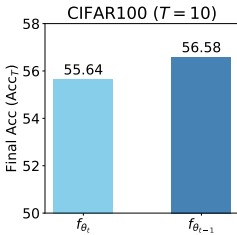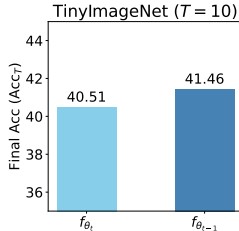

Figure 5: Evaluation of feature extractor for augmented inputs. We evaluate the two variants of the feature extractor for encoding the augmented images in contrastive learning on 10-task splits of CIFAR100 and TinyImageNet datasets.

## F  EFFECT OF PROJECTION HEAD ARCHITECTURE

We analyze how the projection head architecture influences the performance of EFCIL while keeping the similarity-weighted linear-regression-based prototype calibration fixed. Table 8 shows that introducing even a single linear-layer projection head improves performance compared to the case without a projection head. However, adding batch normalization (Ioffe & Szegedy, 2015) slightly reduces accuracy. Increasing the hidden layer dimensionality generally leads to better results, with the best performance observed for hidden layer scaling $\rho \geq 8$. All reported results are obtained without applying the old class repulsion loss to isolate the effect of the projection head architecture.

Table 8: Effect of projection head architecture on model performance. We evaluate the model performance when using a projection head with a different number of layers, hidden layer width scaler ($\rho$), and batch normalization layer on 10-task splits of CIFAR100 and TinyImageNet datasets.

| Projection head architecture | | | CIFAR100 ($T = 10$) | | TinyImageNet ($T = 10$) | |
|---|---|---|---|---|---|---|
| #Layers | $\rho$ | Batch Norm | $\mathrm{Acc}_T$ | $\overline{\mathrm{Acc}}_{1:T}$ | $\mathrm{Acc}_T$ | $\overline{\mathrm{Acc}}_{1:T}$ |
| 0 | – | – | $51.00 \pm 0.57$ | $63.46 \pm 1.67$ | $39.01 \pm 0.49$ | $50.15 \pm 0.66$ |
| 1 | – | ✗ | $54.62 \pm 0.47$ | $66.18 \pm 0.93$ | $40.31 \pm 0.51$ | $52.73 \pm 0.51$ |
| 2 | 1 | ✗ | $55.05 \pm 0.47$ | $66.53 \pm 1.22$ | $40.84 \pm 0.45$ | $52.88 \pm 0.71$ |
| 2 | 2 | ✗ | $55.26 \pm 0.41$ | $66.61 \pm 1.06$ | $40.96 \pm 0.14$ | $53.23 \pm 0.75$ |
| 2 | 8 | ✗ | $55.63 \pm 0.54$ | $\mathbf{66.98} \pm 1.05$ | $\mathbf{41.10} \pm 0.44$ | $53.28 \pm 0.60$ |
| 2 | 32 | ✗ | $\mathbf{55.75} \pm 0.47$ | $66.74 \pm 0.89$ | $41.09 \pm 0.44$ | $\mathbf{53.58} \pm 0.37$ |
| 2 | 32 | ✓ | $55.16 \pm 0.49$ | $66.32 \pm 0.49$ | $40.68 \pm 0.50$ | $53.19 \pm 0.71$ |

## G    EFFECT OF CONTRASTIVE LOSS ON OLD AND NEW TASK ACCURACY

An incremental learner should acquire new knowledge of the current task for the sake of plasticity and also preserve knowledge from previous tasks for the sake of stability. We further compare the stability-plasticity results with and without contrastive loss training in Figure 6. The results show using contrastive loss achieves better performance on older tasks compared to the case without contrastive loss and our method also achieves increased plasticity. These gains in performance are due to richer class-discriminative features learned from contrastive loss.

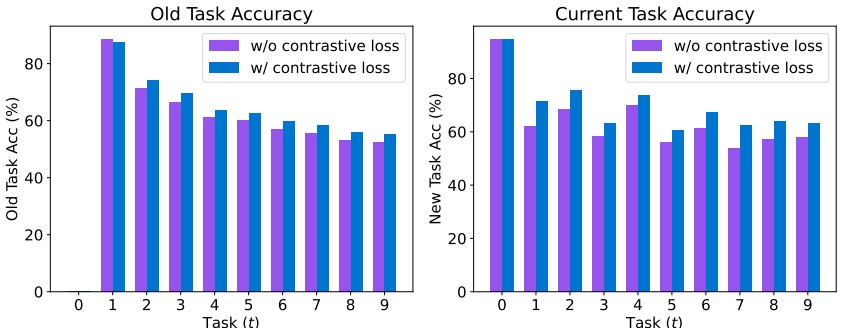

Figure 6: Comparison of old tasks' average accuracy and new task accuracy across training tasks for $T = 10$ on CIFAR100 with and without projector-enhanced contrastive loss.

## H    HYPERPARAMETER EFFECT

The hyper-parameter study in Figure 7 highlights the effect of contrastive loss weight ($\alpha$), contrastive loss temperature ($\tau$), and old class repulsion weight ($\beta$) on the performance of the proposed method for CIFAR100 with $T = 10$ and $T = 20$ tasks. The results indicate that $\alpha$ exhibits a clear trend: performance consistently improves up to $\alpha = 1$, beyond which both final and average incremental accuracies drop sharply, suggesting that overly strong contrastive regularization interferes with class discrimination across new and old tasks in EFCIL. Similarly, $\tau$ demonstrates notable sensitivity, where small values (0.02–0.1) maintain stable accuracy, but larger values ($\tau \geq 0.5$) significantly degrade performance, reflecting the detrimental effect of excessive softening in the contrastive similarity distribution. In contrast, $\beta$ shows a narrow region of stability, with $\beta = 1$ providing performance gains, while higher values cause numerical instability in training due to the covariance matrix becoming singular when new classes are excessively repelled into a constrained subspace. Overall, these results show the performance is obtained around $\alpha = 1$, $\tau = 0.02$–0.1, and $\beta = 1$.

## I    LLM USAGE

We made limited use of LLMs in this research. Specifically, we applied them for grammar checking and correction. In coding, we used them only to obtain code snippets to support visualization. However, we did not use them for problem framing, idea generation, or broader aspects of coding beyond visualization support. .

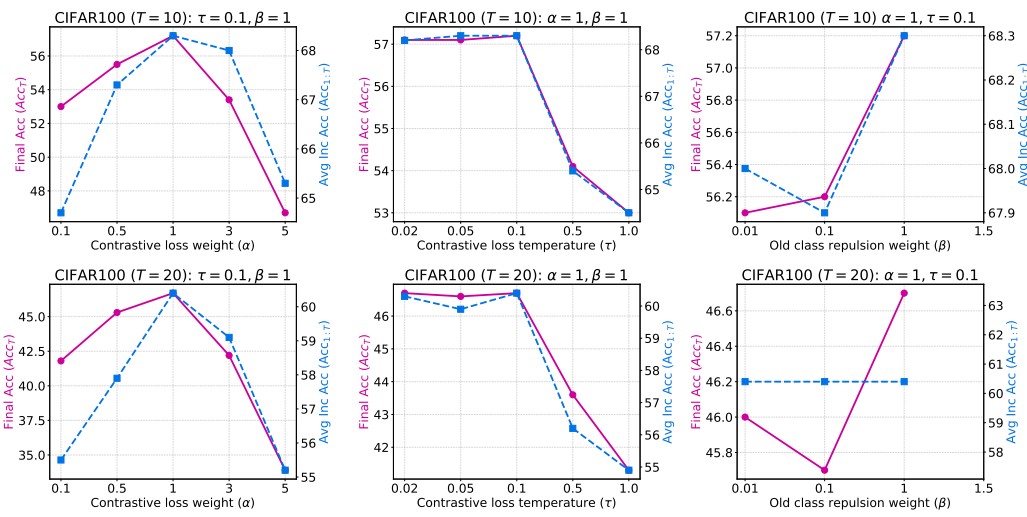

Figure 7: Performances at different settings of hyper-parameters on CIFAR100 datasets.

