# OpenReview forum: "Projection-Enhanced Contrastive Learning and Linear Calibration for Exemplar-Free Class-Incremental Learning"
_ICLR.cc/2026/Conference — Submitted to ICLR 2026_

### Official Review · Reviewer_fpvr · 2025-10-25

**Soundness:** 3
**Presentation:** 3
**Contribution:** 2
**Rating:** 4
**Confidence:** 5

**Summary:**

This paper tackles Exemplar-Free Class-Incremental Learning (EFCIL), proposing a framework that decouples contrastive learning from classification via a projection head, introduces an old-class repulsion loss to reduce feature space congestion, and presents a closed-form similarity-weighted linear regression calibration for prototype adaptation. The method outperforms state-of-the-art baselines on CIFAR100, TinyImageNet, and ImageNet100.

**Strengths:**

1. The use of a projection head to separate contrastive learning from the classification objective is supported by both recent theoretical insights and empirical results.
2. Table 1 and Tables 3-5 demonstrate consistent improvements over recent SOTA, across three benchmark datasets and various task splits. Figure 2 further visualizes the sustained accuracy advantage through incremental tasks.
3. The analysis in Table 2 and Table 8 provides detailed evidence for each component’s contribution.

**Weaknesses:**

1. While the method shows promising results, some core ideas (contrastive learning and prototype calibration) are not entirely new and can be viewed as extensions of existing methods (SDC, SSRE, FCS).
2. Despite referring to theoretical works (Ouyang et al.) supporting the decoupling approach, the paper itself does not provide principled guarantees or deeper theoretical insight into why and when the method will provably outperform single-space approaches. The authors mention this as a limitation but punt the analysis to future work (Section 5). This limits the depth and generality of the contribution.
3. While Figure 7 explores sensitivity to hyperparameters ($\alpha$, $\tau$, $\beta$), there’s only a narrow range, especially for $\beta$. Moreover, the method is sensitive to the contrastive loss weight, $\alpha$.
4. The anonymous link only contained the running command without the code.

**Questions:**

Please refer to Weaknesses

---

> ### Author Response · Authors · 2025-11-21
>
> We thank Reviewer fpvr for their constructive comments on our work.
>
> **Extension of existing methods:** We clarify that our contributions are as follows:
>
> 1. We would like to emphasize that we are the first to identify and address a major limitation of the prior EFCIL works. The limitation is that directly applying contrastive learning in the feature space for classification decreases the performances in cold-start EFCIL settings, as indicated in our ablation study. The proposed solution is to decouple the classification space and contrastive learning space through a projection head.
>
> 2. Our work shows that the contrastive loss can offer the generalization benefit from supervised contrastive loss in EFCIL setting when it’s applied in the post-projected space in terms of decreasing intraclass distance and increasing interclass distance. Along with old class repulsion loss, it reduces feature congestion and is crucial for CIL performance improvement.
>
> 3. We are the first to propose to use asymmetric encoders for two input views; the augmented view is passed through the old task encoder, whereas the normal view is passed through the current task encoder, which provides additional performance increase as observed in Appendix E.
>
> 4. We would like to highlight that another novel contribution comes from developing a swift prototype calibrator that updates full Gaussian prototypes without any significant training costs. In summary, our proposed SLrPC strategy *unifies* the notion of similarity-weighted update from SDC and learnable mapping from LDC into a single closed-form similarity-weighted solution. For each old prototype distribution and new feature, we compute the likelihood of the feature under the prototype distribution and use this as the similarity measure. Importantly, as shown in Appendix C.1, SDC and related methods emerge as *special cases* of our similarity formulation, rather than mere extension. Finally, our similarity-weighted closed-form linear regression can adapt the full prototype distribution using simple matrix updates, thus avoiding the large number of Monte-Carlo samples required by AdaGauss as indicated in line 312.
>
> We will further clarify the above contributions in the camera-ready version.
>
> **Theoretical Explaination:** Ouyang et al. (2025) provide the theoretical foundation for our work and motivate decoupling contrastive learning from the classification feature space via a projection head. Acting as an information bottleneck, the projection head mitigates the inherent tension between contrastive and classification objectives by channeling the strong, augmentation-invariant pressure of the contrastive loss into a separate post-projection space. This prevents the model from distorting the rich, class-discriminative features preserved in the pre-projection (classification) space. In contrast, applying contrastive learning directly on the classification feature space leads to overly compressed representations for new classes due to strong augmentation-invariant nature and also leads to severe distortion of old-class features, which harms the EFCIL performance.
>
>
> **Hyperparameter Range for $\beta$:** We observe that using stronger values for $\beta$ results in a singular covariance matrix resulting in error in model training. Hence, the training occurs in a narrow range of $\beta$ values.
>
>
> **Codebase Unavailability:** We apologize for the inconvenience. The commit rollback didn’t work properly after the bidding, but the source code has now been restored and is available in the anonymous link.

---

### Official Review · Reviewer_Wwyh · 2025-10-29

**Soundness:** 3
**Presentation:** 3
**Contribution:** 3
**Rating:** 6
**Confidence:** 4

**Summary:**

This work introduces a novel exemplar-free class-incremental learning (EFCIL) framework that decouples contrastive learning and classification through a projection head, allowing the model to leverage contrastive objectives without disrupting class-discriminative feature learning. To alleviate feature-space congestion, it further proposes an old-class repulsion strategy operating in the pre-projection space, enhancing separation between old and new classes. Moreover, an efficient prototype calibration method based on closed-form similarity-weighted linear regression is developed to update class prototypes without heavy computation. Together, these innovations improve representation quality and efficiency in exemplar-free continual learning.

**Strengths:**

1. This paper fully analyzes the limitations of existing methods and has a clear research motivation。
2. The proposed closed-form linear-regression-based prototype calibration method demonstrates effectiveness without requiring additional training.
3. The authors demonstrate the effectiveness of the proposed method through a wealth of experiments, and not just bring about performance improvements.

**Weaknesses:**

1. Compared to the previous use of projection-based contrastive learning, what is the innovation of this paper?
2. This paper uses AdaGauss as the baseline. How are the proposed multiple loss functions combined with the original AdaGauss loss?
3. Providing detailed information on the accuracy curve in Figure 2 in the appendix will be beneficial for future research.

**Questions:**

Minor concerns:

- There is no executable code in the provided anonymous repository.

---

> ### Author Response · Authors · 2025-11-21
>
> We thank Reviewer Wwyh for their constructive comments on our work.
>
> **Paper’s Innovation:** We clarify that our contributions as follows:
>
> 1. We would like to emphasize that we are the first to identify and address a major limitation of the prior EFCIL works. The limitation is that directly applying contrastive learning in the feature space for classification decreases the performances in cold-start EFCIL settings, as indicated in our ablation study. The proposed solution is to decouple the classification space and contrastive learning space through a projection head.
>
> 2. Our work shows that the contrastive loss can offer the generalization benefit from supervised contrastive loss in EFCIL setting when it’s applied in the post-projected space in terms of decreasing intraclass distance and increasing interclass distance. Along with old class repulsion loss, it reduces feature congestion and is crucial for CIL performance improvement.
>
> 3. We are the first to propose to use asymmetric encoders for two input views; the augmented view is passed through the old task encoder, whereas the normal view is passed through the current task encoder, which provides additional performance increase as observed in Appendix E.
>
> 4. We would like to highlight that another novel contribution comes from developing a swift prototype calibrator that updates full Gaussian prototypes without any significant training costs. In summary, our proposed SLrPC strategy *unifies* the notion of similarity-weighted update from SDC and learnable mapping from LDC into a single closed-form similarity-weighted solution. For each old prototype distribution and new feature, we compute the likelihood of the feature under the prototype distribution and use this as the similarity measure. Importantly, as shown in Appendix C.1, SDC and related methods emerge as *special cases* of our similarity formulation, rather than mere extension. Finally, our similarity-weighted closed-form linear regression can adapt the full prototype distribution using simple matrix updates, thus avoiding the large number of Monte-Carlo samples required by AdaGauss as indicated in line 312.
>
> We will further clarify the above contributions in the camera-ready version.
>
> **Combination with AdaGauss Loss**:  We use the combined loss $L_t = L_{CE} + L_{AC} + \alpha L_{Con} + \beta L_{OldRep} + \gamma L_{KD} $ defined in line 278. Out of them, we add contrastive training loss $L_{Con}$ and old class repulsion loss $ L_{OldRep}$.
>
> **Information on Accuracy curve in Figure 2:** We have provided information regarding hyper-parameters and source code to generate Figure 2 in Appendix B.
>
> **Codebase Unavailability:** We apologize for the inconvenience. The commit rollback didn’t work properly after the bidding, but the source code has now been restored and is available in the anonymous link.
> ​

---

> ### Comment · Reviewer_Wwyh · 2025-11-25
> **Response to Rebuttal**
>
> Thank you for your response. I'm curious about the reasons behind the performance improvement achieved by using asymmetric encoders for the two input views?

---

> > ### Author Response · Authors · 2025-11-30
> >
> > Thank you Reviewer Wwyh for following up on our previous response.
> >
> >
> > **Reason for improvement from asymmetric encoders**: Using an asymmetric encoder provides additional knowledge-distillation signals during training. In our setup, the contrastive learning objective matches the current-task features produced by $ f_{\theta_t} $ with the corresponding augmented-view features extracted by the previous task’s encoder $ f_{\theta_{t-1}} $. Since the features generated by $ f_{\theta_{t-1}} $ lie in the same representation space as the old-class prototypes, this naturally allows us to incorporate samples from old class prototypes as negatives into the contrastive loss. This can further allow separation between new-class and old-class representations in the post-projection space, thereby helping to reduce feature congestion.

---

### Official Review · Reviewer_NNmi · 2025-10-31

**Soundness:** 2
**Presentation:** 3
**Contribution:** 1
**Rating:** 2
**Confidence:** 5

**Summary:**

This paper proposes a framework for exemplar-free class-incremental learning (EFCIL) that integrates three components: (1) a projection head to decouple contrastive and classification losses, (2) an old-class repulsion loss to reduce feature-space congestion, and (3) a closed-form similarity-weighted linear regression for prototype calibration. Experiments on CIFAR100, TinyImageNet, and ImageNet100 show improved results over existing baselines such as AdaGauss, FCS, and EFC.

**Strengths:**

1. The paper is clearly written and follows the standard EFCIL formulation with comprehensive experiments on multiple benchmarks.

2. The proposed closed-form linear regression calibrator is computationally efficient and practically valuable.

3. Ablation studies are detailed and help to isolate the contributions of each module.

**Weaknesses:**

1. Novelty. Using contrastive learning (PASS) together with prototype estimation (SDC) is already common in EFCIL. This paper appears to make only minor modifications, so the technical contribution is limited. Moreover, inserting a projection head after the backbone and then applying a contrastive loss is standard in SimCLR, SupCon, BYOL/SimSiam, etc.; therefore I remain skeptical that “using a projection layer to separate contrastive and classification losses” constitutes a substantive methodological contribution here.

2. Questionable evaluation focus (cold-start only).
In EFCIL, it is typical to start from a pretrained model or to use half-class as the base task. The paper emphasizes a cold-start protocol (training from scratch) as its primary benchmark, which is unusual given practical usage where a pretrained backbone (e.g., ImageNet/CLIP) is readily available. More importantly, the generalization capability provided by pretraining is crucial for subsequent continual tasks, and thus highly relevant to realistic deployments; it is unclear why the authors restrict attention to this single protocol.

3. Potentially unfair comparisons.
Many compared methods (PASS, IL2A, SSRE, FeTrIL, FeCAM) were originally designed under half-class pretraining/warm-start, so poor cold-start results are expected. Though EFC and AdaGauss report cold-start results in their original works; they also provide warm-start results. I strongly recommend adding warm-start comparisons to ensure fairness and practical relevance

**Questions:**

1. see weakness. My major concerns are novelty and setting.
2. Since the proposed linear-regression calibrator assumes a linear mapping, does it generalize well when feature drift is non-linear?
3. Please provide wall-clock training time and memory comparisons to better demonstrate the claimed “efficiency.”

---

> ### Author Response · Authors · 2025-11-21
>
> We thank Reviewer NNmi for their constructive comments on our work.
>
> **Novelty:** We clarify that our contributions are as follows:
> 1. We would like to emphasize that we are the first to identify and address a major limitation of the prior EFCIL works. The limitation is that directly applying contrastive learning in the feature space for classification decreases the performances in cold-start EFCIL settings, as indicated in our ablation study. The proposed solution is to decouple the classification space and contrastive learning space through a projection head.
>
> 2. Our work shows that the contrastive loss can offer the generalization benefit from supervised contrastive loss in EFCIL setting when it’s applied in the post-projected space in terms of decreasing intraclass distance and increasing interclass distance. Along with old class repulsion loss, it reduces feature congestion and is crucial for CIL performance improvement.
>
> 3. We are the first to propose to use asymmetric encoders for two input views; the augmented view is passed through the old task encoder, whereas the normal view is passed through the current task encoder, which provides additional performance increase as observed in Appendix E.
>
> 4. We would like to highlight that another novel contribution comes from developing a swift prototype calibrator that updates full Gaussian prototypes without any significant training costs. In summary, our proposed SLrPC strategy *unifies* the notion of similarity-weighted update from SDC and learnable mapping from LDC into a single closed-form similarity-weighted solution. For each old prototype distribution and new feature, we compute the likelihood of the feature under the prototype distribution and use this as the similarity measure. Importantly, as shown in Appendix C.1, SDC and related methods emerge as *special cases* of our similarity formulation, rather than mere extension. Finally, our similarity-weighted closed-form linear regression can adapt the full prototype distribution using simple matrix updates, thus avoiding the large number of Monte-Carlo samples required by AdaGauss as indicated in line 312.
>
> We will further clarify the above contributions in the camera-ready version.
>
> **Warm-start Comparisons:** Although warm-start protocols (i.e., training on 50% of the classes before the incremental tasks) are relatively common in EFCIL literature, in many practical scenarios, new tasks come from entirely different distributions. In such cases, pretrained models offer limited utility because the new data can be fundamentally different from existing datasets and one cannot rely on a large base dataset to learn a strong feature extractor beforehand. Thus, cold-start EFCIL becomes more relevant and attracts more and more attention.
>
> Moreover, cold-start EFCIL is generally considered more challenging than warm-start EFCIL. Methods that freeze the feature extractor after a large base task typically perform well in warm-start EFCIL but are ineffective in cold-start cases. Our contributions specifically target this more challenging cold-start regime, which is why our primary evaluation focuses on cold-start results.
>
> Following the reviewer’s suggestion, we conducted additional experiments on warm-start settings. Our method, without hyper-parameter tuning, attains better significant performances on CIFAR100 than all the warm-start baselines, while we have slightly worse performance than a few warm-start methods on ImageNet100. The warm-start methods like FeTrIL and FeCAM achieve better performance because the feature extractor is frozen after the base task and thus retains higher performance for the first half of the classes, boosting the final average incremental accuracy.
>
> Table 1: We report final accuracy and average incremental accuracy on $T=5$ and $T=10$ settings for CIFAR100 and ImageNet100 following AdaGauss. We conducted experiments for IL2A and FCS, and report the remaining results as provided in AdaGauss.
>
> |Method|CIFAR100 (T=5)|CIFAR100 (T=10)|ImageNet100 (T=5) | ImageNet100 (T=10)|
> |--|--|--|--|--|
> |PASS | 54.5 \| 61.8 | 53.8 \| 60.9 | 57.9 \| 64.4 | 58.2 \| 61.8 |
> |IL2A | 54.2 \| 66.4 | 44.3 \| 60.1 | 47.2 \| 57.7 | 47.2 \| 52.3 |
> |SSRE | 55.7 \| 63.9 | 54.9 \| 63.2 | 58.3 \| 65.2 | 61.4 \| 67.7 |
> |FeTrIL | 58.3 \| 65.1 | 56.2 \| 64.6 | 65.6 \| 72.8 | 65.3 \| 72.1 |
> |FeCAM | 60.2 \| 67.2 | 59.9 \| 66.9 | 67.3 \| 75.3 | 67.6 \| 74.9 |
> |FCS | 62.2 \| 70.3 | 61.1 \| 69.1 | 61.8 \| 71.1 | 61.8 \| 70.7 |
> |AdaGauss | 58.9 \| 65.7 | 55.4 \| 63.7 | 66.8 \| 74.1 | 62.8 \| 68.0 |
> |Ours | 64.2 \| 70.8 | 64.9 \| 69.8 | 64.8 \| 72.0 | 59.9 \| 68.9 |

---

> > ### Author Response · Authors · 2025-11-21
> >
> > **Linear-regression Calibration**: If the feature extractor changes extensively, linear mapping cannot address the problem. However, with feature distillation between old and new feature extractors, the prototype change can become moderately non-linear. And, the results show that linear mapping works well in practice. Prior works like FCS also show consistent performance gains when using a single-layer MLP as a calibrator compared to an MLP with multiple layers.
> >
> > **Training Time and Memory**: Please, refer to the wall-clock training time and memory provided in Appendix D for the comparisons.

---

> ### Comment · Reviewer_NNmi · 2025-11-25
>
> Thank you for providing the additional experimental results. However, since the paper relies on the common combination of contrastive learning and prototype correction, I still feel the novelty of the work is somewhat limited. Given the new experimental results, I raise my score to 4 correspondingly.

---

> > ### Author Response · Authors · 2025-11-30
> >
> > We thank the reviewer for your thoughtful followup and for raising the score. We would like to respectfully clarify that our goal is not to present contrastive learning and prototype calibration as new techniques for EFCIL. Rather, our contributions focus on identifying and analyzing a specific failure mode that arises when contrastive learning and classification operate in the same feature space. We show the empirical effectiveness from reducing feature congestion with projection-decoupled contrastive learning and old class repulsion. In addition, we propose similarity-weighted linear calibration for efficient and lightweight prototype calibration.
> >
> >
> > We appreciate the reviewer’s constructive feedback and time spent in reviewing our work.

---

### Official Review · Reviewer_232d · 2025-11-01

**Soundness:** 2
**Presentation:** 3
**Contribution:** 2
**Rating:** 2
**Confidence:** 3

**Summary:**

This paper focuses on the cold start class-incremental learning (CIL). It presents a new exemplar-free CIL approach based on AdaGauss. To reduce feature space congestion, it proposes a contrastive learning approach along with the old-class repulsion strategy. It uses a weighted linear regression to calibrate the prototype to address the prototype drift. The paper claims that the proposed method outperforms previous state-of-the-art methods across multiple EFCIL benchmarks.

**Strengths:**

1. This paper focuses on cold-start CIL, a challenging problem in continual learning.
2. The proposed method achieves competitive performance when compared with previous SOTA methods.
3. Most of the reported data in experiments are averaged over multiple experiments, varying the class order.

**Weaknesses:**

1. The contribution of this paper is incremental. For example,
    1. Introducing contrastive learning into incremental learning is a plug-and-play and effective trick for improving performance, which has been discussed in previous papers (e.g., [3, 4, 5]). The motivation behind introducing contrastive learning in this article is more like improving performance rather than solving problems in CIL.
    2. The use of the least squares technique to solve the problem of prototype offset has been discussed in DPCR [2]. This paper lacks a comparison with DPCR.
2. The projector-enhanced contrastive learning module may probably improve the performance by enhancing the generalization ability of the backbone network instead of addressing forgetting. Improving the supervised image classification task with contrastive learning has been widely discussed [1]. This is a common technique applicable to all image classification problems, but it is not specifically designed for CIL.
3. The claim that the proposed method outperforms existing state-of-the-art methods is not reliable, as it lacks comparison with recent works (e.g., DPCR [2], which is also a method evolving feature extractor with prototype calibration) published in 2025.
4. The source code is NOT available at the provided link.

[1] Islam, Ashraful, et al. "A Broad Study on the Transferability of Visual Representations with Contrastive Learning." *2021 IEEE/CVF International Conference on Computer Vision (ICCV)*. IEEE, 2021.

[2] He, Run, et al. "Semantic Shift Estimation via Dual-Projection and Classifier Reconstruction for Exemplar-Free Class-Incremental Learning." *Proceedings of the 42nd International Conference on Machine Learning*. PMLR, 2025.

[3] Zhu, Jitao, et al. "Class incremental learning with deep contrastive learning and attention distillation." *IEEE Signal Processing Letters* 31 (2024): 1224-1228.

[4] Li, Qiwei, Yuxin Peng, and Jiahuan Zhou. "FCS: Feature calibration and separation for non-exemplar class incremental learning." *Proceedings of the IEEE/CVF conference on computer vision and pattern recognition*. 2024.

[5] He, Run, et al. "REAL: Representation enhanced analytic learning for exemplar-free class-incremental learning." *arXiv preprint arXiv:2403.13522* (2024).

**Questions:**

My key concerns about this paper are listed in the Weaknesses section. Besides, I list some of my concerns that do not impact my rating:

1. What does Table 4.3 in line 408 refer to?
2. Can you provide an analysis of forgetting in the paper (e.g., a comparison between the proposed method and other methods in avoiding forgetting, an analysis of the effectiveness of different modules in avoiding forgetting)?

---

> ### Author Response · Authors · 2025-11-21
>
> We thank Reviewer 232d for their constructive comments on our work.
>
> **Projected-Contrastive Learning Contributions:** Our contributions on contrastive learning part are in 3 folds:
>
> 1. We would like to emphasize that we are the first to identify and address a major limitation of prior EFCIL work. The limitation is that directly applying contrastive learning in the feature space for classification decreases the performances in cold-start EFCIL settings, as indicated in our ablation study. The proposed solution is to decouple the classification space and contrastive learning space through a projection head.
>
> 2. As the reviewer points out, one of the benefits of including contrastive learning is the improvement in generalization. However, our work shows that the contrastive loss can offer such benefits only when it’s applied in the post-projected space in terms of decreasing intraclass distance and increasing interclass distance. It reduces feature congestion and is crucial for CIL performance improvement. Furthermore, using negative samples from prototype distributions in contrastive loss provides complementary signals for reducing feature congestion.
>
> 3. We are the first to propose to use asymmetric encoders for two input views; the augmented view is passed through the old task encoder, whereas the normal view is passed through the current task encoder, which provides additional performance increase as observed in Appendix E.
>
> We will further clarify the above contributions in the camera-ready version.
>
> **Comparison with DPCR:** While DPCR also uses a closed-form linear regression solution and obtains class-specific calibrators by applying SVD on uncentered covariance matrices, our approach follows a different strategy using similarity between class prototypes and new class features.
>
> We ran DPCR on the five seed settings we used for class order, and report the results below. The results in Table 1, 2 and 3 show that our method achieves superior performance across most evaluation settings. On CIFAR100 and TinyImageNet, our approach outperforms DPCR in both final accuracy and average incremental accuracy for T=10 and T=20. On ImageNet100, while DPCR attains a slightly higher final accuracy at T=10 by a small margin 0.02%, our method outperforms it on final accuracy, including both metrics at T=20. Overall, the results confirm that, even when compared against this recent and relevant method, our approach achieves state-of-the-art performance in most evaluation settings in exemplar-free class-incremental learning.
> ​
>
> Table 1: Comparison of average incremental accuracy and final accuracy on the CIFAR100 dataset.
>
> | Methods | $ \text{Acc}_T \, (T{=}10) $ | $ \text{Acc}_T \, (T{=}20) $ | $ \overline{\text{Acc}}_{1:T} \, (T{=}10) $ | $ \overline{\text{Acc}}_{1:T} \, (T{=}20) $ |
> |---|---|---|---|---|
> | DPCR | 49.94 | 39.97 | 61.92 | 53.98 |
> | **Ours** | **56.58** | **46.87** | **67.54** | **59.39** |
> ​
>
> Table 2: Comparison of average incremental accuracy and final accuracy on TinyImageNet dataset.
>
> | Methods | $ \text{Acc}_T \, (T{=}10) $ | $ \text{Acc}_T \, (T{=}20) $ | $ \overline{\text{Acc}}_{1:T} \, (T{=}10) $ | $ \overline{\text{Acc}}_{1:T} \, (T{=}20) $ |
> |---|---|---|---|---|
> | DPCR | 35.63 | 27.05 | 48.31 | 41.65 |
> | **Ours** | **41.46** | **34.64** | **53.71** | **47.72** |
> ​
>
> Table 3: Comparison of average incremental accuracy and final accuracy on the ImageNet100 dataset.
>
> | Methods | $ \text{Acc}_T \, (T{=}10) $ | $ \text{Acc}_T \, (T{=}20) $ | $ \overline{\text{Acc}}_{1:T} \, (T{=}10) $ | $ \overline{\text{Acc}}_{1:T} \, (T{=}20) $ |
> |---|---|---|---|---|
> | DPCR | **53.32** | 41.36 | 67.12 | 57.93 |
> | **Ours** | 53.30 | **44.13** | **67.86** | **60.09** |
> ​
>
> Furthermore, we experimented replacing our SLrPC calibrator with DPCR while keeping other components intact, and report the results in Table 4. The results show that our method still outperforms the DPCR setup.
>
> Table 4: Comparison of average incremental accuracy and final accuracy on the CIFAR100 dataset by replacing SLrPC with DPCR.
>
> | Methods | $ \text{Acc}_T \, (T{=}10) $ | $ \text{Acc}_T \, (T{=}20) $ | $ \overline{\text{Acc}}_{1:T} \, (T{=}10) $ | $ \overline{\text{Acc}}_{1:T} \, (T{=}20) $ |
> |---|---|---|---|---|
> | Ours - SLrPC + DPCR | 44.89 |55.58 |  57.9 |67.11 |
> | **Ours** | **46.87**| **56.58** | **59.39** | **67.54**|
> ​
>
> **Codebase Unavailability:** We apologize for the inconvenience. The commit rollback didn’t work properly after the bidding, but the source code has now been restored and is available in the anonymous link.

---

> ### Author Response · Authors · 2025-11-21
>
> **Table 4.3 in line 408:** The reference in line 408 was intended for Table 2 and not for section 4.3 and we will correct it in the updated version.
>
> **Forgetting Analysis:** We provide forgetting results compared to EFC and AdaGauss in the table below. The results in Table 4 show that AdaGauss and our method observe a decrease in forgetting compared to EFC. Our method achieves only comparable performance compared to Adagauss, because the initial performance is comparatively higher in our method. Table 5 further illustrates that this pattern arises from the use of contrastive learning and the old-class repulsion loss: while these components improve task performance by reducing feature congestion between old and new classes, the higher initial accuracy results in a larger absolute forgetting value. However, we still achieve better average incremental accuracy and final accuracy.
> ​
> Table 4: Forgetting analysis on CIFAR100-10, TinyImageNet, and ImageNet (lower is better).
>
> | Method   | CIFAR100-10 | TinyImageNet | ImageNet |
> |----------|-------------|--------------|----------|
> | EFC      | 28.27       | 31.42        | 34.56    |
> | Adagauss | 13.64       | 16.34        | 16.66    |
> | Ours     | 15.82       | 18.44        | 20.78    |
> ​
>
> Table 5: Ablation on CIFAR100-10 showing forgetting measured under different training components (lower is better).
>
> | Setting | Accuracy |
> |-------------|----------|
> | LR only                                      | 12.17    |
> | LR + Contrastive only                        | 15.26    |
> | LR + Contrastive only + Old class Rep (Ours) | 15.82    |

---

> ### Comment · Reviewer_232d · 2025-11-26
> **Reject: A Well-Engineered Framework with Limited Scope and Impact for Continual Learning**
>
> Thank you to the authors for their timely and professional responses to my questions. However, my fundamental concerns regarding the contribution (Weakness 1) and the motivation (Weakness 2) remain largely unresolved. Key reasons are as follows:
>
> 1. The authors' rebuttal confirms that the performance gain comes largely from a significant boost in the initial accuracy of each task instead of reducing forgetting. While improving plasticity is valuable, the central challenge in continual learning is the stability-plasticity trade-off. The presented results suggest that the method primarily enhances plasticity, with the reduction in forgetting being a relative byproduct of a stronger starting point, rather than a direct outcome of a novel mechanism designed to address catastrophic forgetting. It feels more like a representation learning technique that is applied to a continual learning setting, rather than a fundamental advance in continual learning theory or mechanism.
> 2. Reason 1 puts this paper in a dilemma: (a) if the technique proposed by the author are highly inspiring, this method should be presented as a contrastive learning paper; (b) if the technique is limited to only one continual learning method or cannot inspire future continual learning approaches, the innovation of the paper obviously cannot meet the standards of ICLR.
> 3. Of course, the proposed method involves more than just contrastive learning. But as I mentioned in Weakness 1, the proposed method is solid engineering work involving a bag of tricks, but it lacks enough distinguished contributions. More importantly, the inspiration of these techniques may be limited due to being difficult to transfer to other continual learning methods, thus limiting the impact of the paper.
> 4. The authors claim that a contribution is to be the first to propose to use asymmetric encoders for two input views. However, I did not see the description of how the asymmetric encoders work in the main text or in the pseudo-code. A key contribution should be fully discussed in the main text. Besides, due to the lack of sufficient information, I am not able to determine whether the asymmetric encoders are sufficiently different from previous work (e.g., PRAKA [1]).
>
> Overall,  the contributions feel more like a proficient refinement of existing ideas, falling short in demonstrating the requisite level of novelty and breadth of impact. Therefore, I still tend to reject this paper.
>
> [1] Shi, Wuxuan, and Mang Ye. "Prototype reminiscence and augmented asymmetric knowledge aggregation for non-exemplar class-incremental learning." *Proceedings of the IEEE/CVF International Conference on Computer Vision*. 2023.

---

> ### Author Response · Authors · 2025-12-01
>
> We thank the reviewer 232d for following up on our response.
>
> **Enhancement of plasticity and stability:** We would like to respectfully clarify that, although our contrastive formulation does increase initial accuracy, we also observe performance improvements for old task accuracies. To provide a more direct view of stability, we compare old task accuracy and new task accuracy in Appendix G (Figure 6) with and without projected-contrastive loss. Furthermore, for details, we provide a comparison with more cases in this [link](https://anonymous.4open.science/r/iclr2026-D134/old-and-new_taskacc.pdf). The results show that old-class accuracy with projected contrastive learning is consistently better in both cases without contrastive loss and when contrastive learning is applied to the classification space. Furthermore, our old class repulsion loss further elevates old class task performance by reducing feature congestion between tasks. Thus, the gains cannot be attributed solely to a boosted starting point, and our method maintains relatively higher old class performances.
>
>
> **Applicability beyond a single continual learning method:** We applied the projected contrastive learning loss on top of the LwF + prototype drift compensator [1] on CIFAR100 and TinyImagenet datasets. We observe performance improvement compared to the original method as observed in the table below.
>
>
> |Method| $\overline{\mathrm{Acc}}_{1:T}$| $\overline{\mathrm{Acc}}_{1:T}$|
> |---|---|---|
> |LwF + Drift compensator|59.50|46.80|
> |LwF + Drift compensator + contrastive |61.69|47.95|
>
>
> **Asymmetric Encoding and Difference from PRAKA:** We encode the augmented views through the old feature extractor $ f_{\theta_{t-1}} $. This creates a form of task-temporal asymmetry that encourages the evolving encoder to remain closer to earlier representations, offering some stability across tasks. We have discussed asymmetric encoding in line 238 in the main text and highlighted it in Figure 1. We agree that this point can be emphasized more clearly and will revise the main text accordingly.
>
> Our asymmetric encoding is fundamentally different from asymmetric knowledge aggregation in PRAKA. While our idea is focused on the feature space regularization, PRAKA’s asymmetric knowledge aggregation (AKA) is focused on the classifier. AKA is targeted to transfer the knowledge learned from 4-way rotations-based self-supervised learning classifiers’ ensemble to a single classifier. The asymmetry in PRAKA arises because the old class logits are obtained from augmented prototypes whereas the new-task logits are averaged across the four rotation augmentations for new-task samples. In contrast, our idea is to apply asymmetry in the feature encoding level. We apply two separate feature extractors for the different views (old feature extractor and evolving feature extractor). For this reason two asymmetric strategies serve different purposes and operate at different stages of the model.
>
> We appreciate the reviewer’s constructive feedback and time spent in reviewing our work.
>
> [1] Alex Gomez-Villa, Dipam Goswami, Kai Wang, Andrew D. Bagdanov, Bartlomiej Twardowski, & Joost van de Weĳer (2024). Exemplar-Free Continual Representation Learning via Learnable Drift Compensation. In ECCV (7) (pp. 473-490).

---

### Meta-Review · Area_Chair_woip · 2026-01-07

**Summary:**

This paper studies the exemplar-free class-incremental learning problem. To address this problem, the authors propose a novel framework that decouples contrastive learning and classification via a projection head, enabling contrastive learning while preserving rich class-discriminative features in the pre-projection space. An anonymous link for the code was provided, but it had expired on January 6, 2026. The paper was reviewed by four expert reviewers: three of them were negative towards it, while one reviewer gave a borderline acceptance. There are some important concerns raised by the reviewers, such as 1) the novelty issue, as the method is a common combination of contrastive learning and prototype correction, and 2) the projector-enhanced contrastive learning module may probably improve the performance by enhancing the generalization ability of the backbone network instead of addressing forgetting. The authors partially addressed the concerns, but the major concerns above were not fully addressed. Most of the reviewers tend not to upgrade their ratings. Therefore, I tend to reject this paper for now. The authors are encouraged to revise the paper based on the reviewers' comments and resubmit it to the next venue.

**Reviewer Concerns:**

The paper was reviewed by four expert reviewers: three of them were negative towards it, while one reviewer gave a borderline acceptance. There are some important concerns raised by the reviewers, such as 1) the novelty issue, as the method is a common combination of contrastive learning and prototype correction, and 2) the projector-enhanced contrastive learning module may probably improve the performance by enhancing the generalization ability of the backbone network instead of addressing forgetting. The authors partially addressed the concerns, but the major concerns above were not fully addressed.

**Reviewer Scores:**

The paper was reviewed by four expert reviewers: three of them were negative towards it, while one reviewer gave a borderline acceptance. The authors partially addressed the concerns, but the major concerns above were not fully addressed. Most of the reviewers tend not to upgrade their ratings.

---

### Decision · Program_Chairs · 2026-01-26

Reject